# SLIT3 fragments orchestrate neurovascular expansion and thermogenesis in brown adipose tissue

Tamires Duarte Afonso Serdan[1], Heidi Cervantes[1,13], Benjamin Frank[1,13], Akhil Gargey Iragavarapu [1], Qiyu Tian[1], Daniel Hope[1], Chan Hee J. Choi[2], Anne Hoffmann [3], Adhideb Ghosh [4], Christian Wolfrum [4], Matthew B. Greenblatt [5], Paul Cohen [2], Matthias Blüher [3,6], Halil Aydin[1,7,8], Gary J. Schwartz [9,10] & Farnaz Shamsi [1,7,11,12] ✉

Brown adipose tissue is an evolutionary innovation in placental mammals that regulates body temperature through adaptive thermogenesis. Cold exposure activates brown adipose tissue thermogenesis through coordinated induction of brown adipogenesis, angiogenesis, and sympathetic innervation; however, how these processes are coordinated remains unclear. Here, we show that fragments of Slit guidance ligand 3 (SLIT3) drive crosstalk among adipocyte progenitors, endothelial cells, and sympathetic nerves. Adipocyte progenitors secrete SLIT3, which is cleaved into functionally distinct SLIT3-N and SLIT3-C fragments that independently promote angiogenesis and sympathetic innervation. We identify PLXNA1 as a receptor for SLIT3-C and demonstrate its essential role in sympathetic innervation of brown adipose tissue. Moreover, we identify BMP1 as the first SLIT protease described in vertebrates. Coordinated neurovascular expansion mediated by distinct SLIT3 fragments provides a bifurcated yet integrated mechanism that ensures a synchronized brown adipose tissue response to environmental challenges. Finally, this study reveals a previously unrecognized role for adipocyte progenitors in regulating tissue innervation.

The regulation of body temperature is a fundamental homeostatic process in endothermic animals. Preserving a stable internal temperature ensures the efficiency and fidelity of all cellular reactions and is essential for survival. Brown adipose tissue (BAT) is a specialized type of adipose tissue that is primarily responsible for regulating body temperature through adaptive thermogenesis. Brown adipocytes oxidize substrates and generate heat to maintain euthermia in a cold environment[1].

Brown adipocytes are embedded within an intricate network of blood vessels and sympathetic nerves that support their development and thermogenic function. The high thermogenic activity of BAT requires a high rate of blood perfusion to supply $O_2$ and substrates.

The sympathetic nerves also play a key role in stimulating BAT thermogenesis by releasing norepinephrine in the tissue. Norepinephrine activates adrenergic signaling in thermogenic adipocytes, resulting in enhanced thermogenesis and lipolysis[2]. Chronic cold exposure increases BAT mass by de novo recruitment of brown adipocytes, as well as by expanding the network of blood vessels and sympathetic nerves in the tissue. This coordinated expansion of BAT ensures its continuous responsiveness to hormonal and neuronal stimuli and is essential for enhanced thermogenesis in cold[3,4]. However, how these distinct processes are spatiotemporally coordinated remains unclear. Furthermore, while significant progress has been made in understanding the molecular mechanisms of thermogenic activation in

adipocytes, the equally crucial process of remodeling the thermogenic adipose niche remains less understood.

Cell–cell communication is vital for organismal development and function. Ligand-receptor interactions allow cells in complex tissues to coordinate their functions during development, homeostasis, and remodeling. To understand the mechanisms through which different cell types in BAT communicate and synchronize their response to cold to collectively enhance thermogenesis, we previously used single-cell transcriptomic data[5] to construct a network of ligand-receptor interactions in BAT[6]. That study highlighted the central role of adipocyte progenitors, not merely as the source of adipocytes, but also as key communicators and versatile players in the adipose tissue microenvironment. These progenitors have been shown to contribute to a variety of processes, including extracellular matrix remodeling, immune modulation, and angiogenesis[7].

The SLIT family of secreted proteins and their roundabout (ROBO) receptors constitute a conserved signaling pathway originally identified for its role in axon guidance during neurodevelopment[8–10]. SLIT guidance ligands are large extracellular glycoproteins that regulate the migration and adhesion of multiple cell types including neurons, endothelial cells, leukocytes, and epithelial cells[11]. Studies in Drosophila and mice revealed the role of Slit proteins in neuronal axon guidance and arborization, vascularization and angiogenesis, inflammatory cell chemotaxis, and tumor metastasis[12,13]. In mammals, the Slit family consists of three members: SLIT1 is predominantly expressed in the nervous system whereas SLIT2 and SLIT3 are also expressed outside the nervous system[13–16]. Although originally identified as axon guidance factors, SLITs regulate the migration and positioning of cells in various tissues. However, the mechanisms controlling the context-dependent outcomes of SLIT signaling are still largely unclear.

Here, we demonstrate that SLIT3 is a critical regulator of BAT neurovascular development and function. We show that SLIT3 mediates crosstalk amongst adipocyte progenitors, endothelial cells, and sympathetic nerves, regulating both angiogenesis and innervation. Using BAT- and adipocyte progenitor-specific loss-of-function models, we show that the loss of SLIT3 disrupts both angiogenesis and sympathetic innervation, ultimately blunting cold-induced BAT thermogenesis. We identify bone morphogenetic protein 1 (BMP1) as the protease responsible for SLIT3 cleavage and reveal that the resulting fragments activate distinct pathways: SLIT3-N promotes angiogenesis, while SLIT3-C, through its receptor PLXNA1, drives sympathetic innervation. This pathway demonstrates a sophisticated level of intercellular coordination whereby a single factor simultaneously drives these two distinct processes essential for thermogenic activation.

## Results
### Identification of SLIT3-ROBO4 signaling axis in BAT
Intercellular communication plays a critical role in coordinating tissue adaptation to external challenges. To define the role of intercellular crosstalk in cold-induced BAT thermogenesis, we previously used single-cell transcriptomic data to build a network of ligand-receptor interactions involved in the cold-induced remodeling of BAT[6]. This analysis revealed the significance of adipocyte progenitors as the major communication hub in the adipose niche[6]. This prompted us to search for ligands secreted from adipocyte progenitors that might mediate the crosstalk between adipocyte progenitors and their niche. Ligand-receptor analysis identified SLIT3 and its putative endothelial receptor, ROBO4, as a potential crosstalk axis in BAT. In BAT, *Slit3* is predominantly expressed in *Pdgfra*+ and *Trpv1*+ adipocyte progenitors, as well as in vascular smooth muscle cells, while its receptor, *Robo4*, is exclusively expressed in vascular and lymphatic endothelial cells (Fig. 1a). To validate these findings, we separated the stromal vascular and adipocyte fractions of mouse BAT, sorted distinct cell populations, and measured *Slit3* expression. This analysis confirmed that adipocyte progenitors are the primary source of *Slit3* in BAT under

both basal and cold-acclimated conditions (Fig. 1b and Supplementary Fig. 1a–d), whereas *Robo4* is exclusively expressed in endothelial cells (Supplementary Fig. 1e). To examine SLIT3 protein expression in BAT and its regulation by environmental temperature, we performed Western blot analysis on BAT from mice housed at different temperatures. Using an antibody that detects the N-terminal region of SLIT3 protein, we found that SLIT3-FL levels were elevated in BAT of mice exposed to cold for 2 days compared to those maintained at room temperature (Fig. 1c, d). These findings indicate that cold exposure enhances SLIT3 protein levels in BAT.

A previous study reported that SLIT3 is secreted from macrophages in white adipose tissue (WAT)[17]. However, our unbiased scRNA-seq data, along with targeted expression analysis in isolated macrophages, did not detect any *Slit3* expression in BAT macrophages (Fig. 1a, b). Analysis of scRNA-seq datasets from mouse inguinal white adipose tissue (ingWAT) and perigonadal white adipose tissue (pgWAT), as well as human WAT[18], revealed *SLIT3* expression in adipocyte progenitors, mesothelial cells, and mural cells (vascular smooth muscle cells and pericytes) (Supplementary Fig. 2a–d), but no expression in macrophages in either mouse or human WAT (Supplementary Fig. 2a–d). Collectively, these findings suggest that adipocyte progenitors, rather than macrophages, are the primary source of *SLIT3* in adipose tissue.

To assess the temporal pattern of Slit3 expression during adipogenesis, we measured SLIT3 protein levels in a mouse brown adipocyte progenitor cell line at multiple time points throughout in vitro adipogenic differentiation. SLIT3 protein levels sharply decreased by days 4 and 8 of differentiation, in contrast to the increasing uncoupling protein 1 (UCP1) expression (Supplementary Fig. 2e). These results indicate that SLIT3 expression is significantly higher in adipocyte progenitors than in mature adipocytes, where SLIT3 protein levels were minimal.

### SLIT3 is essential for BAT thermogenesis
To investigate *Slit3* function in BAT, we used AAV-mediated shRNA delivery to knock down *Slit3* expression specifically in BAT. We confirmed a significant reduction in *Slit3* transcript and protein levels in BAT, but not in WAT, of mice receiving *Slit3* shRNAs (Supplementary Fig. 3a–d). Mice lacking *Slit3* expression in BAT exhibited severe impairments in cold-induced thermogenesis, maintaining lower core body temperature (Fig. 1e) and BAT temperature (Supplementary Fig. 3e) during cold exposure.

After 7 days of cold exposure, BAT analysis revealed pronounced whitening and increased lipid accumulation (Fig. 1f), accompanied by reduced expression of *Ucp1* (Fig. 1g–i), iodothyronine deiodinase type 2 (*Dio2*), cell death-inducing DFFA-like effector A (*Cidea*), and beta-3 adrenergic receptor (*Adrb3*), as well as mitochondrial, and lipolysis-related genes in *Slit3* knockdown mice (Fig. 1i). Furthermore, *Slit3* loss resulted in a marked reduction in mitochondrial electron transport chain complex levels (Fig. 1j, k). Mice lacking *Slit3* expression in BAT showed a comparable reduction in *Ucp1* transcript and protein levels when housed at room temperature (Supplementary Fig. 4a–c) or under thermoneutral conditions (30 °C; Supplementary Fig. 4d–f). However, the expression of other thermogenic and mitochondrial genes was only modestly reduced under these conditions (Supplementary Fig. 4a, d).

Consistent with histological and molecular analyses, *Slit3* loss in BAT significantly reduced energy expenditure, oxygen consumption, and $CO_2$ production in mice housed at 5 °C (Fig. 1l, m and Supplementary Fig. 5). Importantly, these metabolic alterations occurred without changes in body weight, body composition, BAT and WAT depot weights, respiratory exchange ratio, food intake, or locomotor activity (Supplementary Fig. 5). Together, these findings establish SLIT3 as a critical regulator of cold-induced BAT thermogenesis and energy expenditure.

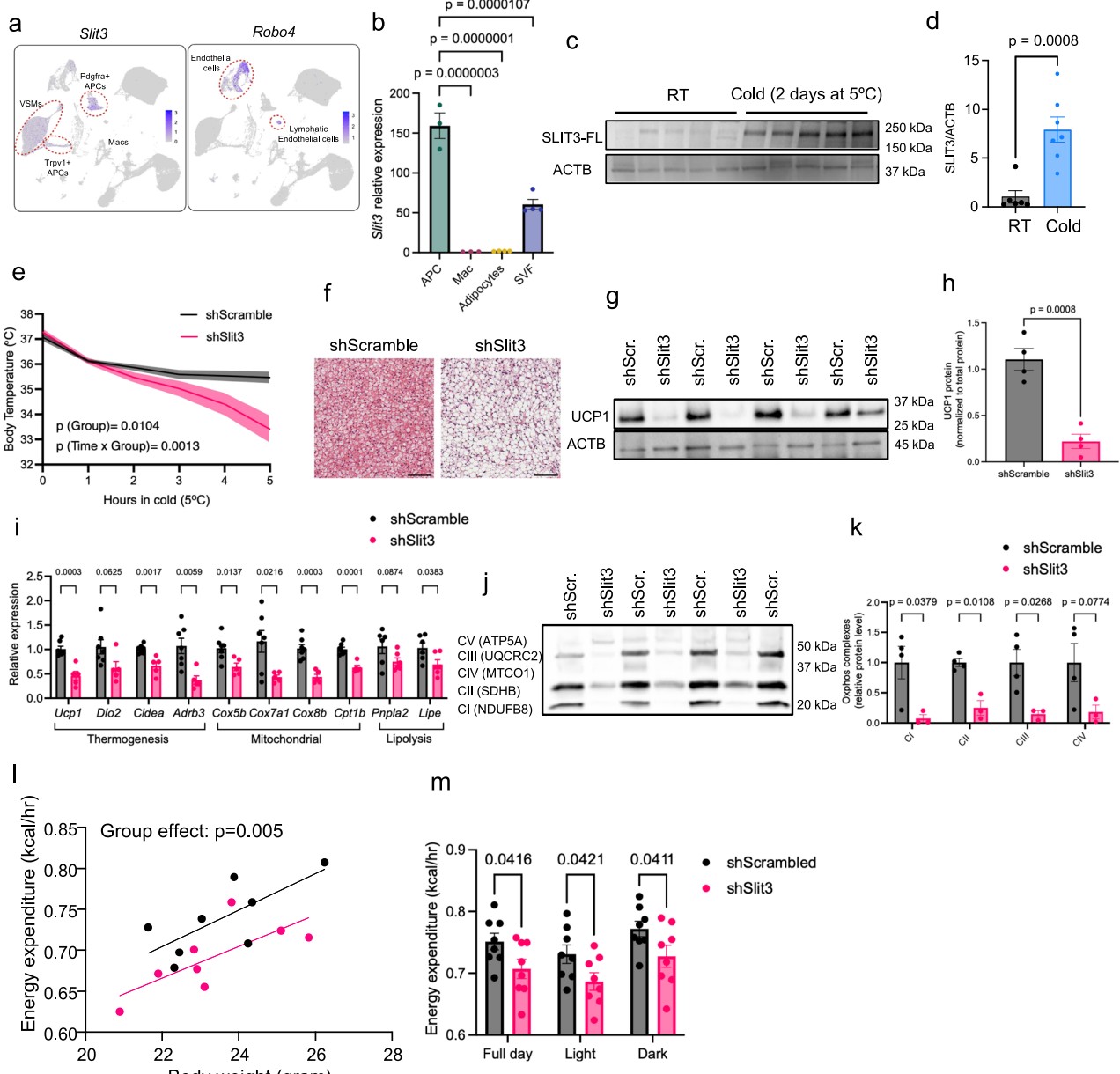

**Fig. 1 | Axon Guidance Molecule Slit3 is essential for BAT thermogenesis. a** Slit3 and Robo4 expression in scRNA-seq data from mouse BAT. **b** Slit3 expression in isolated adipocyte progenitors, macrophages, adipocytes, and the total stromal vascular fraction from mouse BAT. $n = 3, 3, 4$, and 4 samples. Each sample was derived from a pool of three animals. **c, d** SLIT3 protein levels in BAT of mice housed at room temperature (RT) or cold (5 °C for 2 days). $n = 6, 7$ animals. **e** Cold tolerance test in AAV-shScramble or AAV-shSlit3-injected mice. $n = 13, 12$ mice. **f** Representative Hematoxylin and Eosin (H&E) staining of BAT. Scale bar = 100 μm. Data were reproduced in four animals per group. **g, h** UCP1 protein level in BAT of mice after 7 days at cold (5 °C). $n = 4$ animals per group. **i** Expression of thermogenic, mitochondrial, and lipolysis related genes in BAT of mice after 7 days at cold

(5 °C). $n = 7, 5$ animals. **j, k** OxPhos complex protein levels in BAT of mice receiving AAV-shSlit3 or scramble shRNA and housed at cold (5 °C) for 7 days. $n = 4, 3$ animals. **l** Regression plots of energy expenditure versus total body mass in AAV-shSlit3 or AAV-shScramble-injected mice housed at 5 °C. $n = 8$ animals per group. **m** Average daily energy expenditure in AAV-shSlit3 or AAV-shScramble-injected mice housed at 5 °C. $n = 8$ animals per group. Data are presented as means ± SEM and analyzed by one-way ANOVA with Dunnett's multiple comparison test (**b**), two-way ANOVA repeated measures (**e**), unpaired two-sided Student's $t$ tests (**d, h, i, k**), ANCOVA (**l**), and two-way ANOVA (**m**). The experiments were repeated three times with similar results. Source data are provided in the Source Data file.

## Loss of SLIT3 impairs BAT neurovasculature

To investigate how the loss of SLIT3 impairs cold-induced BAT thermogenesis, we first examined the cell-autonomous effects of SLIT3 on adipocytes. We treated in vitro-differentiated brown adipocytes with recombinant SLIT3-N and SLIT3-C proteins. qPCR and immunoblot analyses showed no changes in the expression of *Ucp1* or other thermogenic genes in adipocytes treated with recombinant SLIT3 fragments (Supplementary Fig. 6a, b). These results led us to hypothesize

that SLIT3 loss impairs BAT thermogenesis by disrupting the brown adipocyte niche.

The expansion of BAT neurovasculature is essential for cold adaptation and enhanced thermogenesis in the cold[3,4]. Chronic cold exposure significantly increases the density of parenchymal sympathetic neurites and capillary blood vessels in BAT (Supplementary Fig. 6c–h). To investigate the mechanism(s) underlying impaired BAT thermogenesis in *Slit3*-deficient mice, we assessed vascularization and

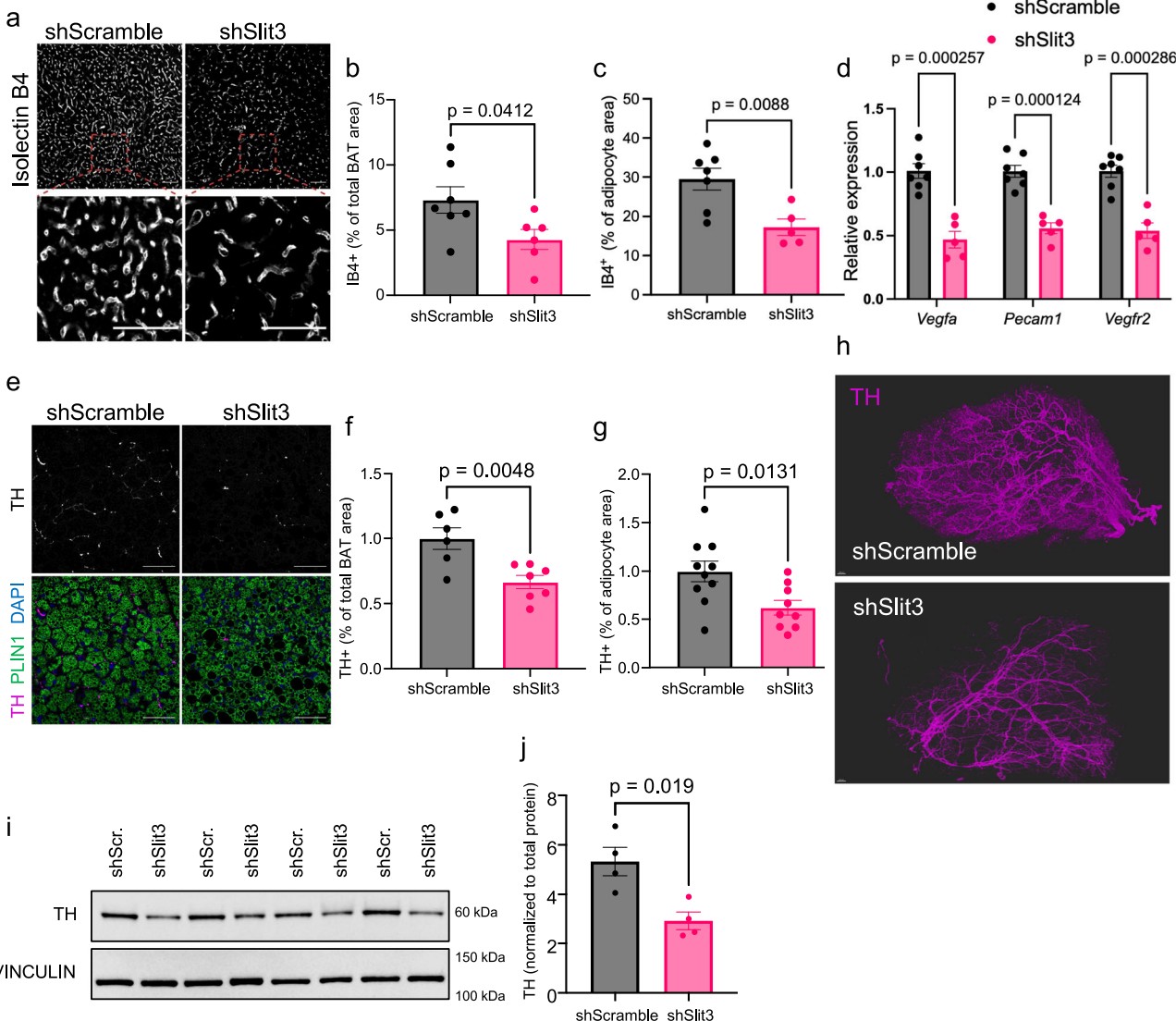

**Fig. 2 | Loss of Slit3 impairs cold-induced angiogenesis and sympathetic innervation in BAT. a** Representative images of Isolectin B4 (IB4) staining in BAT of mice receiving AAV-shSlit3 or scramble shRNA after 7 days at cold (5 °C). Scale bar = 50 μm. **b, c** Quantification of the percentage of Isolectin B4 (IB4)$^+$ area normalized to the total tissue area. $n$ = 7, 6 animals. **b** or to the PLIN1$^+$ area. $n$ = 7, 5 animals, **c** in BAT from mice treated with AAV-shSlit3 or scramble shRNA after 7 days of cold exposure (5 °C). **d** Expression of endothelial cell markers in BAT of mice receiving AAV-shSlit3 or scramble shRNA after 7 days at cold (5 °C). $n$ = 7, 5 animals. **e** Representative images of TH and PLIN1 staining in BAT of mice receiving AAV-shSlit3 or scramble shRNA after 7 days at cold (5 °C). Scale bar = 50 μm.

**f, g** Quantification of the percentage of TH$^+$ area normalized to the total tissue area. $n$ = 6, 7 animals, **f** or to the PLIN1$^+$ area. $n$ = 10, 9 animals, **g** in BAT of mice treated with AAV-shSlit3 or scramble shRNA after 7 days of cold exposure (5 °C). **h** 3D reconstruction of TH staining in BAT of mice receiving AAV-shSlit3 or scramble shRNA after 7 days at cold (5 °C). **i, j** TH protein levels in BAT of mice receiving AAV-shSlit3 or scramble shRNA after 7 days at cold (5 °C). $n$ = 4 animals per group. Data are presented as means ± SEM and analyzed by unpaired two-sided Student's $t$ tests. The experiments were repeated three times with similar results. Source data are provided in the Source Data file.

sympathetic innervation in BAT from *Slit3* knockdown or control mice after 7 days of cold exposure (5 °C). Analysis of BAT vasculature revealed that *Slit3* knockdown significantly impaired angiogenesis, as indicated by a reduction in capillary density (Fig. 2a–c) and a decrease in endothelial-specific transcript expression (Fig. 2d).

The loss of *Slit3* also resulted in a dramatic reduction in the density of tyrosine hydroxylase (TH)-expressing neurites in BAT (Fig. 2e–g). To assess the architecture and density of sympathetic innervation, we used the Adipo-Clear protocol[19] to stain TH-expressing sympathetic nerves in BAT, followed by lightsheet microscopy. 3D volumetric images of TH-positive nerves revealed a marked decline in parenchymal innervation in *Slit3*-deficient BAT (Fig. 2h). Furthermore, total TH protein levels in BAT were significantly reduced in the absence of SLIT3 (Fig. 2i, j). Notably, even when animals were housed at room

temperature or thermoneutrality, loss of SLIT3 led to a decrease in both capillary density and sympathetic innervation in BAT (Supplementary Fig. 7a–l). However, housing at cold temperatures further exacerbated these defects. These findings underscore the crucial role of SLIT3 in the development and expansion of the neurovascular network in BAT, especially during adaptation to cold environments.

To directly test whether impaired sympathetic innervation underlies the thermogenic defects in *Slit3*-deficient BAT, we bypassed neural input by administering the β3-adrenergic receptor agonist CL-316,243. *Slit3* knockdown animals treated with CL-316,243 for 10 days exhibited fully intact induction of thermogenic, mitochondrial, and lipolytic gene programs (Supplementary Fig. 7m). These results demonstrate that the intrinsic ability of brown adipocytes to respond to adrenergic stimulation remains uncompromised in the absence of

SLIT3. Thus, the primary role of SLIT3 in BAT is not to modulate adipocyte-intrinsic thermogenic capacity, but rather to drive the proper neurovascular expansion necessary for cold-induced thermogenesis.

### ASPC-derived SLIT3 promotes neurovascular expansion in cold

To further confirm that adipocyte stem and progenitor cells (ASPCs) are the primary source of SLIT3 and that ASPC-derived SLIT3 is essential for cold-induced neurovascular expansion and BAT thermogenesis, we specifically deleted *Slit3* in *Pdgfra*-expressing cells by crossing *Slit3* floxed mice[20] with an inducible *Pdgfra*-cre strain[21] (*Pdgfra*-creERT2;*Slit3*^flox/flox^ or *Slit3*^iΔAPC^). Western blotting showed a complete loss of SLIT3 protein in BAT and a marked reduction in WAT depots of *Slit3*^iΔAPC^ mice (Fig. 3a–h and supplementary Fig. 8a–d). Consistent with the observed phenotypes in BAT-specific SLIT3 knockdown mice, both male and female *Slit3*^iΔAPC^ mice exhibited significant impairments in BAT thermogenesis and were cold-intolerant (Fig. 3i, j), highlighting the essential role of adipocyte progenitor-derived SLIT3 in cold adaptation. H&E analysis of the BAT from *Slit3*^iΔAPC^ and control littermates revealed no difference in lipid droplet size (Fig. 3k). However, consistent with our findings from Slit3 knockdown experiments, *Slit3*^iΔAPC^ mice exposed to cold exhibited a significant reduction in both sympathetic neurite density and capillary density in BAT (Fig. 3l–o). Together, these results confirm that *Pdgfra*-expressing adipocyte progenitors are the primary source of SLIT3 in BAT, and that the loss of ASPC-derived SLIT3 disrupts cold-induced neurovascular expansion, leading to impaired thermogenic activation.

### BMP1-mediated SLIT3 cleavage produces two ligands

Members of the SLIT family are proteolytically cleaved into a large N-terminal and a shorter C-terminal fragment[9]. However, the functional implications of SLIT cleavage and the distinct roles played by the full-length protein and its fragments remain unclear. Additionally, the SLIT proteases in vertebrates have not been identified.

To investigate whether SLIT3 is cleaved in adipocyte progenitors, we overexpressed an N- and C-terminal tagged wild-type SLIT3 (SNAP-SLIT3-HaloTag) as well as an uncleavable SLIT3 variant (SNAP-SLIT3UC-HaloTag) in a brown adipocyte progenitor cell line (Fig. 4a). Using antibodies against HaloTag and SNAP-tag, we detected the full-length SLIT3 protein (SLIT3-FL) in the total cell lysates of cells overexpressing either the wild-type or uncleavable SLIT3 constructs (Fig. 4b, c). The HaloTag antibody detected two bands in the conditioned media of cells overexpressing SNAP-SLIT3-HaloTag: a ~200 kDa band corresponding to the tagged full-length SLIT3 and a ~60 kDa band matching the expected size of the tagged SLIT3 C-terminal fragment (SLIT3-C). In contrast, the media from cells expressing the uncleavable SLIT3 only contained the SLIT3-FL (Fig. 4b). Similarly, the SNAP-tag antibody detected both the SLIT3-FL and a ~150 kDa band corresponding to the tagged SLIT3 N-terminal fragment (SLIT3-N) in the media of cells overexpressing SNAP-SLIT3-HaloTag (Fig. 4c). A minor amount of SLIT3-N, but not SLIT3-C, was also detected in the total cell lysates, which contain membrane and membrane-associated proteins (Fig. 4c). Consistent with prior reports that SLIT-N can remain membrane-associated while Slit-C is more diffusible[9], we observed SLIT3-C exclusively in the media and SLIT3-N in both lysates and media (Fig. 4c), supporting a model in which fragment properties enable a differential range of action in BAT.

Moreover, analysis of a proteomics dataset of the adipocyte secretome showed the presence of SLIT3 in the conditioned media of in vitro differentiated adipocytes derived from mouse visceral (Visc), subcutaneous inguinal white (SubQ), and interscapular BAT[22] (Fig. 4d). SLIT3 was significantly more abundant in the conditioned media of brown and subcutaneous white adipocytes than visceral white adipocytes. These results indicated that SLIT3 is secreted by various adipocyte types in vitro, with the highest abundance observed in the secretome of brown adipocytes.

A previous study suggested a role for the metalloprotease tolkin (TOK) in SLIT proteolysis in *Drosophila melanogaster*[23]. TOK is a member of the BMP1/Tolloid-like family of metalloproteases. In mammals, the BMP1/Tolloid-like family includes bone morphogenetic protein 1 (BMP1), tolloid (TLD), and tolloid-like 1 and 2 (TLL1 and TLL2). Given the similarity in the cleavage sites among the members of the SLIT family[23], we hypothesized that the BMP1/TLD-like proteases could be involved in SLIT3 cleavage. Among the members of the BMP1/TLD-like protease family, only *Bmp1* and *Tll1* are expressed in BAT (Fig. 4e and Supplementary Fig. 9). Notably, *Bmp1* is abundantly expressed in *Pdgfra*- and *Trpv1*-expressing adipocyte progenitors, the major cell types expressing *Slit3* in BAT (Fig. 4e). BMP1 lacks growth factor activity found in other BMP family members and instead functions as a metalloprotease. To discern the role of BMP1 in SLIT3 proteolysis in adipocyte progenitors, we used a combination of pharmacological and genetic loss-of-function studies. We showed that the inhibition of BMP1 activity using a small molecule inhibitor, UK383,367[24], blocked SLIT3 cleavage (Fig. 4f). Similarly, knocking down *Bmp1* using siRNAs prevented SLIT3 cleavage (Fig. 4g). Thus, we concluded that BMP1 is responsible for the proteolytic processing of SLIT3, establishing BMP1 as the SLIT protease in vertebrates.

### SLIT3 fragments promote angiogenesis and innervation in BAT

To investigate the specific functions of SLIT3 fragments in BAT, we generated AAV constructs for adipocyte-specific overexpression of full-length SLIT3 (SLIT3-FL), its N-terminal fragment (SLIT3-N, aa 1–1116), and its C-terminal fragment (SLIT3-C, aa 1117–1523) (Fig. 5a). Four weeks after AAV administration in the interscapular BAT, we confirmed the specific expression of SLIT3-FL, SLIT3-N, and SLIT3-C (Fig. 5b, c). Overexpression of SLIT3-FL and SLIT3-C, but not SLIT3-N, enhanced sympathetic innervation in BAT, as indicated by an increased number and relative area of TH-expressing sympathetic neurites (Fig. 5d–f). Furthermore, overexpression of SLIT3-FL and SLIT3-N led to increased capillary density in BAT (Fig. 5g, h). Mice overexpressing SLIT3-FL and SLIT3-C, but not SLIT3-N, also exhibited higher TH levels in BAT (Fig. 5i–q and Supplementary Fig. 10). Notably, SLIT3-FL and SLIT3-C overexpression enhanced BAT thermogenesis, as evidenced by increased UCP1 expression and higher BAT temperatures under cold exposure (Fig. 5k, n, q, r). These findings indicate that SLIT3 fragments directly promote the expansion of BAT's neurovascular network, with SLIT3-C primarily driving sympathetic innervation and SLIT3-N inducing angiogenesis, collectively contributing to enhanced thermogenic function.

To determine the effects of SLIT3 overexpression in the absence of cold stimulation, we overexpressed SLIT3-FL, SLIT3-N, and SLIT3-C and assessed TH and UCP1 levels in BAT from mice housed at room temperature. Consistent with the results observed in cold, overexpression of SLIT3-FL and SLIT3-C, but not SLIT3-N, increased TH levels in BAT (Supplementary Fig. 11). These effects were, however, more modest at room temperature compared to cold exposure. Moreover, overexpression of SLIT3-FL and SLIT3-C at room temperature did not increase UCP1 protein levels, indicating that while SLIT3-FL and SLIT3-C can directly promote sympathetic innervation under basal conditions, their impact becomes more pronounced during cold exposure, when sympathetic activation and thermogenic demand are elevated (Supplementary Fig. 11).

### PLXNA1 mediates SLIT3-dependent sympathetic innervation

Full-length and N-terminal SLIT fragments bind to transmembrane ROBO receptors (ROBO1-4)[9], while the receptor binding and bioactivity of the C-terminal fragment (SLIT-C) remains less characterized. The C-terminal of SLIT2 has been shown to bind Plexin A1 (PLXNA1)[25],

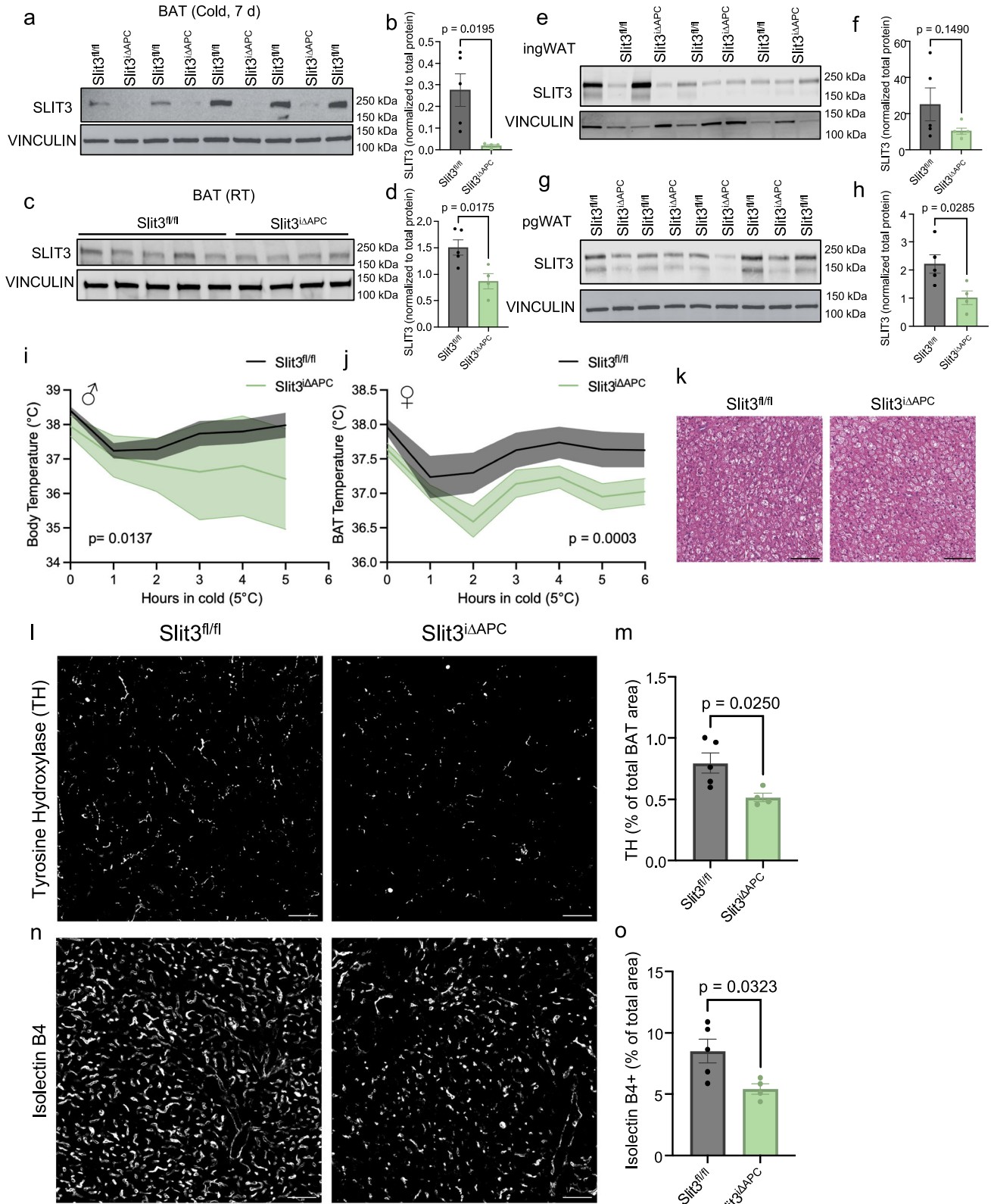

**Fig. 3 | Adipocyte-specific Slit3 deletion impairs cold-induced thermogenesis.** **a–h** SLIT3 protein levels in BAT, ingWAT, and pgWAT depots of Slit3^fl/fl and Slit3^iΔAPC mice. **b–d** $n = 5$, 4 animals. **f** $n = 5$ animals per group. **h** $n = 5$, 4 animals. **i, j** BAT temperature during the cold tolerance test in **i** male. $n = 5$, 4 animals, **j** female Slit3^fl/fl and Slit3^iΔAPC mice. $n = 8$ animals per group. **k** Representative Hematoxylin and Eosin (H&E) staining of BAT. Scale bar = 25 μm. **l** Representative images of TH staining in BAT, **m** quantification of the percentage of TH⁺ area, **n** representative images of Isolectin B4 (IB4) staining, and **o** quantification of the percentage of Isolectin B4 (IB4)⁺ area in BAT of Slit3^fl/fl and Slit3^iΔAPC mice after 7 days at cold (5 °C). Scale bar = 50 μm. $n = 5$, 4 animals. Data are presented as means ± SEM and analyzed by repeated measures ANOVA (**i**, **j**) and unpaired two-sided Student's $t$ tests (**b**, **d**, **f**, **h**, **m**, **o**). Source data are provided in the Source Data file.

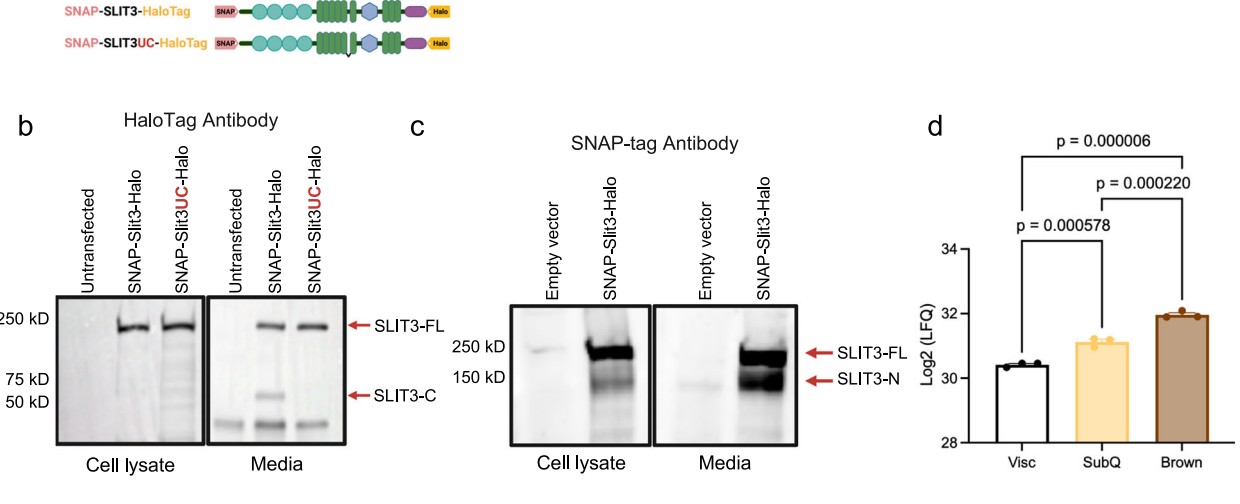

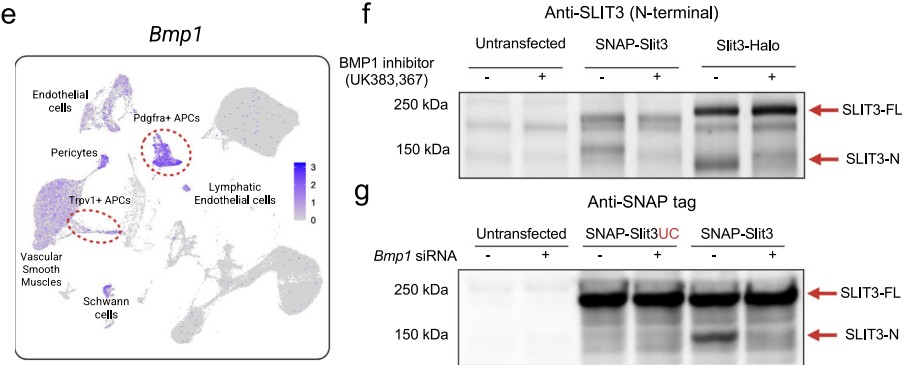

**Fig. 4 | BMP1-mediated proteolytic cleavage of SLIT3 generates two secreted ligands. a** Schematic of the tagged Slit3 transgenes. **b, c** Western blots using HaloTag (**b**) and SNAP-tag (**c**) antibodies to visualize SLIT3 fragments in cell lysates and conditioned media from adipocyte progenitors expressing the indicated plasmids. **d** Quantification of total SLIT3 peptide area in conditioned media from in vitro differentiated primary visceral (Visc), subcutaneous (SubQ), and brown adipocytes by mass spectrometry. $n = 3$ samples per group. **e** Expression of Bmp1 in adipocyte progenitors from scRNA-seq data of mouse BAT. **f** Western blot for SLIT3 in adipocyte progenitors transfected with the indicated constructs and treated with Bmp1 inhibitor UK383,367 or vehicle. **g** Western blot using SNAP-tag antibody in adipocyte progenitors transfected with siBMP1 or scrambled siRNA and over-expressing the indicated constructs. Data are presented as mean ± SEM and analyzed by one-way ANOVA with Tukey's multiple comparisons test (**d**). Source data are provided in the Source Data file.

but the receptor(s) for SLIT3-C have yet to be identified. To identify the receptors mediating SLIT3 signaling in BAT, we analyzed the expression of candidate SLIT receptors. scRNA-seq analysis of BAT revealed that among the four Robo family members, only *Robo4* is expressed in BAT, where it is exclusively localized to vascular and lymphatic endothelial cells (Fig. 1a and Supplementary Fig. 1e). To validate this, we co-stained BAT sections with a ROBO4 antibody and Isolectin B4. ROBO4 was specifically localized to Isolectin B4-labeled capillaries (Fig. 6a). This combined with the observation that SLIT3-N increased capillary density (Fig. 5g, h), supports an endothelial SLIT3-ROBO4 axis. Furthermore, double staining of BAT with ROBO1 and TH antibodies revealed ROBO1 expression in TH-expressing sympathetic neurites (Fig. 6b). Similarly, PLXNA1 and TH were co-localized in sympathetic neurites (Fig. 6c).

Building on our finding that SLIT3-C enhances sympathetic innervation (Fig. 5d–f) and that PLXNA1 is expressed on sympathetic neurites (Fig. 6c), we hypothesized that PLXNA1 serves as the receptor for SLIT3-C. To address this, we first assessed the ability of PLXNA1 to interact with SLIT3-FL or SLIT3-C. We overexpressed the extracellular region of PLXNA1, C-terminally fused to a 6X histidine tag, along with SLIT3-FL-HaloTag or SLIT3-C-HaloTag in a mouse adipocyte progenitor cell line. Using a His-tag antibody, we immunoprecipitated PLXNA1 and its interacting proteins. We found that both SLIT3-FL and SLIT3-C co-immunoprecipitated with the extracellular region of PLXNA1 (Fig. 6d). These findings indicate that PLXNA1 specifically interacts with the C-terminal region of SLIT3 (SLIT3-C), which is distinct from the N-terminal region recognized by ROBO receptors[26].

To further investigate the interaction between PLXNA1 and SLIT3, we used AlphaFold2 Multimer to model complexes involving the extracellular domain of PLXNA1 and the C-terminal region of SLIT3. Multiple structure predictions were generated based on sequence alignments and structural relaxation, with only those having average inter-protein TM (iPTM) + predicted TM (pTM) scores >0.8 retained for further analysis. Diagnostic plots were then used to assess model quality (Supplementary Fig. 12). The top-ranked models, shown in Fig. 6e, f, reveal two distinct interaction modes: a monomeric PLXNA1-SLIT3 complex (Fig. 6e) and a dimeric PLXNA1 interacting with a monomeric SLIT3 (Fig. 6f). The predicted binding interfaces of SLIT3 and SEMA6D, another known ligand for PLXNA1, on PLXNA1's extracellular domains suggest a partial overlap between the SLIT3 and SEMA6D binding regions (Fig. 6g, h). The AlphaFold2 diagnostic plots (Supplementary Fig. 12) further validate the structural integrity of the models, with Predicted Aligned Error (PAE) plots for both the

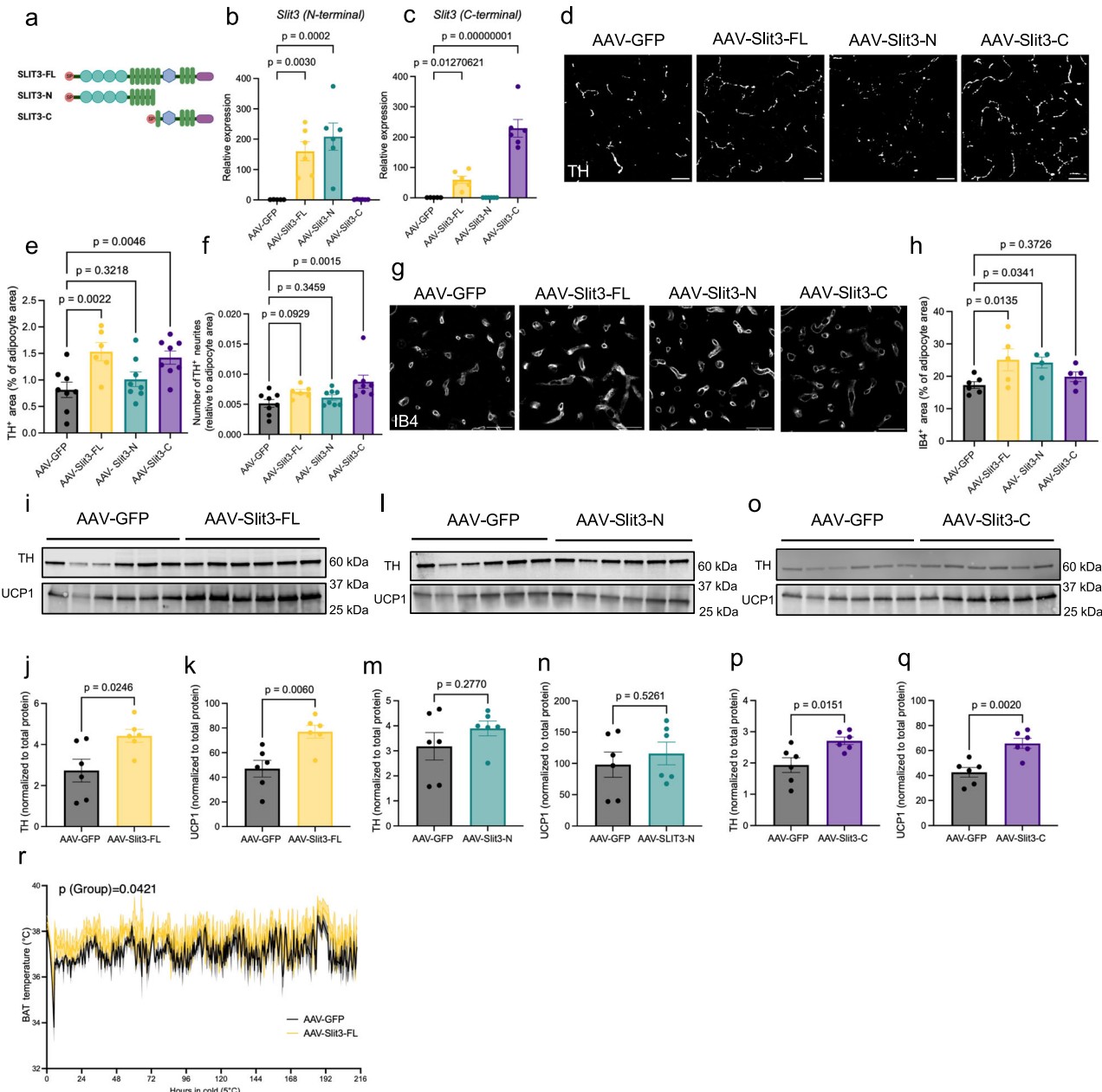

**Fig. 5 | SLIT3 fragments promote angiogenesis and sympathetic innervation in BAT. a** Schematic representation of Slit3-FL, Slit3-N, and Slit3-C constructs. **b**, **c** Expression of Slit3 in BAT measured by qPCR using primers targeting the **b** N-terminal or **c** C-terminal regions of the Slit3 transcript. *n* = 5, 6, 6, and 6 animals. **d** Representative images of TH staining in BAT of mice expressing AAV-GFP, AAV-Slit3-FL, AAV-Slit3-N, or AAV-Slit3-C after 7 days of cold exposure (5 °C). Scale bar = 25 µm. **e**, **f** Quantification of TH⁺ **e** area and **f** neurites normalized to the PLIN1⁺ area in BAT of mice expressing AAV-GFP, AAV-Slit3-FL, AAV-Slit3-N, or AAV-Slit3-C after 7 days of cold exposure (5 °C). *n* = 8, 6, 8, and 8 animals. **g** Representative images of Isolectin B4 staining in BAT from mice expressing AAV-GFP, AAV-Slit3-FL, AAV-Slit3-N, or AAV-Slit3-C after 7 days of cold exposure (5 °C). Scale bar = 50 µm. **h** Quantification of Isolectin B4⁺ capillary area normalized to the PLIN1⁺ area in BAT

of mice expressing AAV-GFP, AAV-Slit3-FL, AAV-Slit3-N, or AAV-Slit3-C after 7 days of cold exposure (5 °C). *n* = 6, 5, 4, and 5 animals. **i**–**q** Western blots and quantification of TH and UCP1 in BAT from mice expressing AAV-GFP, AAV-Slit3-FL (**i**–**k**), AAV-Slit3-N (**l**–**n**), or AAV-Slit3-C (**o**–**q**) after 7 days of cold exposure (5 °C). *n* = 6 animals per group. **r** BAT temperature during cold exposure in mice expressing AAV-GFP or AAV-Slit3-FL. *n* = 3 animals per group. *N* = 6–8 per group. Data are presented as mean ± SEM and analyzed by one-way ANOVA with Dunnett's multiple comparisons test (**b**, **c**, **e**, **f**, **h**), unpaired two-sided Student's *t* test (**j**, **k**, **m**, **n**, **p**, **q**), and two-way ANOVA with a mixed-effects model with Geisser-Greenhouse correction (**r**). The experiments were repeated two times with similar results. Source data are provided in the Source Data file.

monomeric and dimeric complexes, along with predicted Local Distance Difference Test (pLDDT) scores, demonstrating high confidence in the predicted structures.

We next examined whether PLXNA1 functions as a receptor for SLIT3-C in vivo and whether it is required for SLIT3-mediated sympathetic innervation. To test this, we overexpressed SLIT3-C in BAT with or without *Plxna1* knockdown. Consistent with earlier findings (Fig. 5),

SLIT3-FL overexpression significantly enhanced sympathetic innervation in BAT (Fig. 6i, j). However, this effect was diminished when PLXNA1 expression was knocked down (Fig. 6i, j). Similarly, *Plxna1* knockdown markedly reduced the SLIT3-C-induced increase in sympathetic innervation (Fig. 6k, l). Notably, in the absence of PLXNA1, SLIT3-C overexpression failed to elevate BAT temperature in response to cold exposure (Fig. 6m). Mice co-expressing SLIT3-C and shPlxna1 in

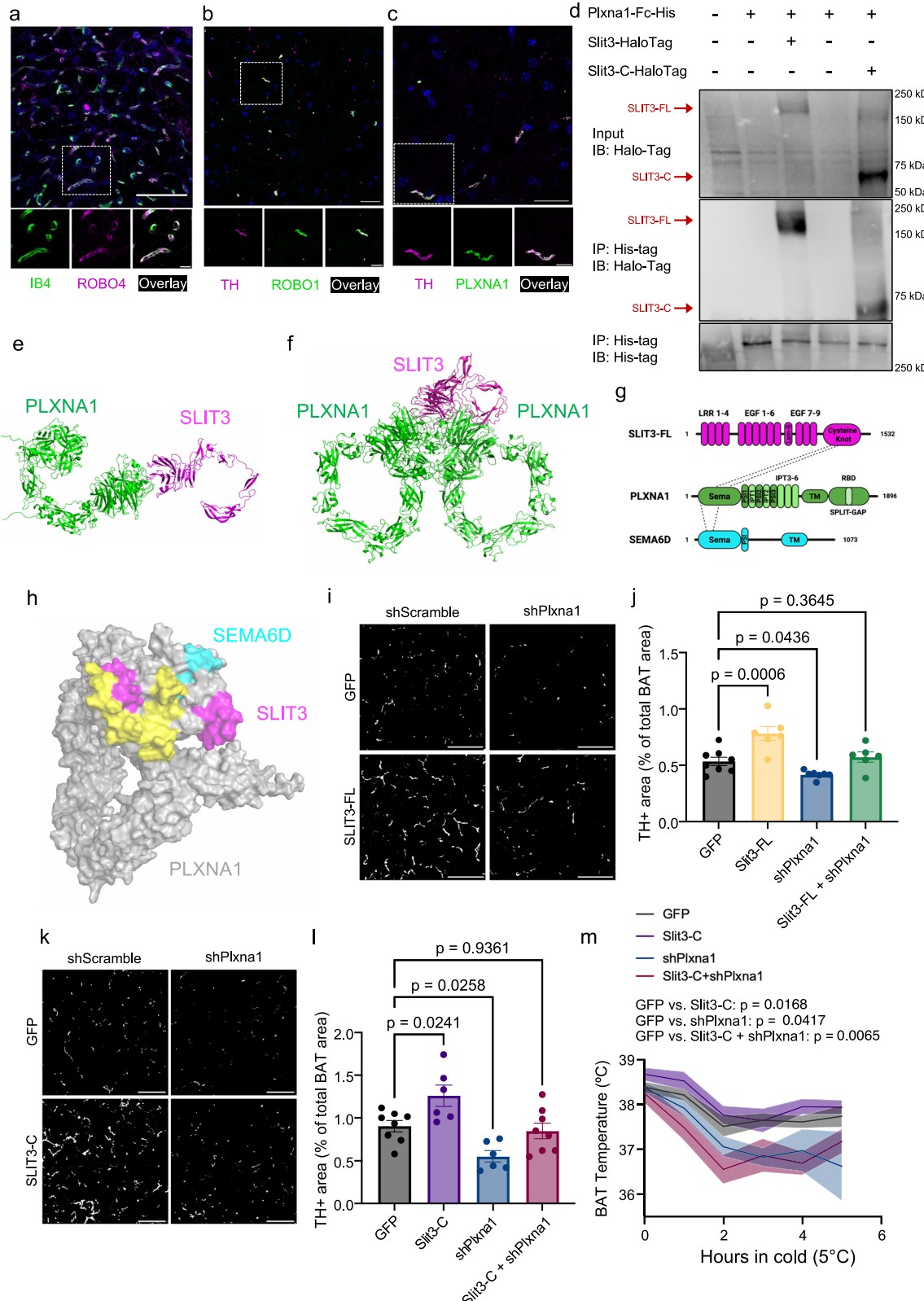

BAT displayed normal levels of sympathetic innervation but showed reduced BAT temperature in cold. We speculate that this effect may reflect additional roles of PLXNA1 in sympathetic neurons beyond axon growth, potentially involving pathways downstream of semaphorin ligands. Collectively, by combining biochemical assays, computational structural modeling, and in vivo studies, we provide strong and converging evidence that PLXNA1 is a direct and essential receptor

for the C-terminal region of SLIT3, required for both sympathetic innervation and thermogenic activation in BAT.

## PLXNA1 is essential for sympathetic innervation in BAT

Given the severe reduction in sympathetic innervation of BAT and cold tolerance in mice receiving Plxna1 shRNAs (shPlxna1 group in Fig. 6i–m), we hypothesized that PLXNA1 is crucial for the

**Fig. 6 | PLXNA1 is the SLIT3 receptor that mediates sympathetic innervation in BAT. a**–**c** Immunofluorescence co-staining in BAT: **a** ROBO4 with Isolectin B4, **b** ROBO1 with TH, and **c** PLXNA1 with TH. Scale bars = 50 μm (**a**), 25 μm (**b**, **c**). **d** Co-immunoprecipitation of His-tagged PLXNA1 with Halo-tagged SLIT3-FL or SLIT3-C. **e**, **f** Ribbon representations of the top-ranked monomeric PLXNA1-SLIT3 (**e**) and dimeric PLXNA1 and monomeric SLIT3 (**f**) interaction models. **g** Domain organization of human SLIT3 (magenta), PLXNA1 (green), and SEMA6D (cyan). Dashed lines indicate binding regions between SLIT3-PLXNA1 and SEMA6D-PLXNA1. Human SLIT3 is comprised of an LRR (leucine-rich repeat) region, EGF-like (epidermal growth factor-like) domains, a LamG-like (Agrin, Laminin, Perlecan and SLIT; ALPS) module, and a C-terminal cysteine knot. SEMA6D consists of Sema and PSI domains in the extracellular region. PLXNA1 contains a Sema domain, three PSI domains, and six IPT (immunoglobulin-like, plexins, and transcription factors) domains extracellularly, while a cytoplasmic Split-GAP domain includes a GAP (GTPase-activating protein) domain interrupted by an RBD (Rho-binding domain). **h** Human SLIT3 and SEMA6D interaction interfaces mapped on the molecular surfaces of PLXNA1 extracellular domains (residues 27–1244) using AlphaFold2 model. The SLIT3 and SEMA6D binding interfaces are shown in magenta and cyan, respectively. The common binding footprint is colored in yellow. **i** Representative images and **j** quantification of TH+ area in BAT of mice expressing Slit3-FL with or without shPlxna1 after 7 days of cold exposure (5 °C). Scale bar = 50 μm. n = 8, 6, 7, and 6 animals. **k** Representative images and **l** quantification of TH+ area in BAT of mice expressing Slit3-C with or without shPlxna1 after 3 days of cold exposure (5 °C). Scale bar = 50 μm. n = 8, 6, 6, and 8 animals. **m** BAT temperature during the cold tolerance test in mice expressing Slit3-C with or without shPlxna1. n = 8 animals per group. Data are presented as mean ± SEM. Statistical analysis was performed using one-way ANOVA with Dunnett's multiple comparisons test (**f**, **h**, **j**, **l**) and repeated measures ANOVA with Dunnett's test (**m**). The experiments were repeated three times with similar results. Source data are provided in the Source Data file.

development and cold-induced expansion of sympathetic neurites in BAT. To directly test this, we injected AAVs delivering Plxna1 shRNAs into BAT. qPCR and western blot analysis confirmed a significant reduction of PLXNA1 expression in BAT (Supplementary Fig. 13a, b). Loss of PLXNA1 led to a pronounced decrease in sympathetic nerve density in mice housed at room temperature as well as after 3 and 7 days of cold exposure (Fig. 7a, b). This was accompanied by a reduction in total TH protein levels and NE concentrations (Fig. 7c–e and Supplementary Fig. 12c, d) in BAT, while capillary density remained unchanged (Fig. 7f, g). This resulted in impaired BAT thermogenesis, evidenced by reduced BAT temperature in cold conditions (Fig. 7h) and a downregulation of the thermogenic gene program both at room temperature and after 3 days of cold acclimation (Fig. 7i, j). Loss of PLXNA1 reduced expression of GAP43, a key axonal growth cone protein involved in axon outgrowth and regeneration. This was reflected by a decreased GAP43+ neurite area and a lower proportion of sympathetic neurites expressing GAP43 (Fig. 7k–m). Consistently, *Gap43* mRNA levels were also reduced in BAT following *Plxna1* knockdown (Fig. 7n). Together, these findings establish PLXNA1 as a key regulator of sympathetic innervation in BAT, demonstrating its essential role in both the steady-state maintenance and cold-induced expansion of sympathetic neurites in BAT (Fig. 7o).

## SLIT3 expression correlates with adipose tissue health
Lastly, to assess the significance of SLIT3 in humans, we measured its expression in adipose tissue biopsies from abdominal subcutaneous WAT and omental visceral WAT, collected from two independent cohorts of the Leipzig Obesity BioBank (LOBB). In the metabolically healthy versus unhealthy obese cohort (MHUO)[27], we found that *SLIT3* transcript levels in human abdominal subcutaneous WAT were positively correlated with serum adiponectin concentrations ($r = 0.546$, $P = 0.00176$, $N = 31$) and negatively regulated with the relative number of macrophages in omental visceral WAT ($r = -0.566$, $P < 0.001$, $N = 31$) in insulin-sensitive patients (Fig. 8a, b), but not in insulin-resistant patients (Fig. 8c, d).

Additionally, we examined *SLIT3* expression in the human cross-sectional cohort composed of paired samples of omental visceral and abdominal subcutaneous WAT from 1480 individuals. In this cohort, we found *SLIT3* expression in omental visceral WAT negatively correlated with Hemoglobin A1C (HbA1c) ($r = -0.092$, $P < 0.001$, $N = 1447$) (Fig. 8e). Additionally, *SLIT3* expression in omental visceral and abdominal subcutaneous WAT was positively associated with circulating level of the anti-inflammatory adipokine Omentin1 ($r = 0.164$, $P < 0.001$, $N = 675$ and $r = 0.124$, $P = 0.00121$, $N = 675$, respectively) (Fig. 8f, g). Omentin1 has been shown to play important roles in glucose homeostasis, lipid metabolism, insulin resistance, and diabetes[28,29]. Collectively, these data suggest that *SLIT3* may regulate adipose tissue health and inflammation in humans, potentially impacting insulin sensitivity.

## Discussion
Recent advances in technical and computational methods for studying complex tissues are providing a systematic understanding of how cellular crosstalk within the tissue microenvironment drives development, function, and disease mechanisms. Using scRNA-seq data from BAT, this study identifies SLIT3 fragments as niche factors mediating crosstalk amongst adipocyte progenitors, sympathetic nerves, and blood vessels. We show that loss of SLIT3 or PLXNA1 in BAT significantly impairs cold-induced neurovascular expansion and adaptive thermogenesis. Our work introduces the concept that adipocyte progenitors are dynamic regulators of adipose tissue remodeling, expanding the view of their role beyond adipogenesis alone. We uncover molecularly specific enzymatic and receptor-mediated mechanisms that coordinate angiogenesis and sympathetic innervation, two essential processes for maintaining adipose tissue health (Fig. 7o).

A bifurcated signaling system enables independent yet coordinated regulation of vascular and neuronal components, providing greater flexibility and temporal precision in adapting to environmental stimuli. In the context of SLIT3, proteolytic processing introduces an additional level of regulation, as the full-length protein and its fragments exhibit distinct cell association and diffusion properties. Notably, SLIT3-C is exclusively detected in the media and not in the cell lysate, suggesting enhanced diffusibility and a capacity to mediate long-range effects on sympathetic innervation, whereas SLIT3-FL and SLIT3-N are more likely to function in a localized manner (Fig. 4c)[9].

Despite the critical roles of SLIT proteins in many processes, the biological activities of SLIT fragments and the mechanisms controlling the context-dependent outcomes of SLIT signaling are still largely unclear. This is in part due to the unknown identity of the SLIT proteases, which prevented the separation of SLIT fragment activities from that of the full-length protein. Identification of SLIT proteases and the specific receptors for the SLIT fragments are critical for understanding the mechanisms that control SLIT signaling during tissue development, remodeling, and regeneration. In this work, we identified the metalloprotease BMP1 as the vertebrate SLIT protease. Through a combination of pharmacological and genetic approaches, we found that BMP1 is responsible for SLIT3 cleavage. This finding highlights BMP1's role in regulating the biological activities of SLIT3, and potentially other SLIT family members, in various contexts, including axon guidance and growth, angiogenesis, inflammatory cell chemotaxis, and tumor metastasis.

BMP1 is a key protease that processes several extracellular matrix proteins and growth factors. It cleaves substrates such as procollagen, lysyl oxidase, and proteoglycans, which are essential for tissue integrity, including in adipose tissue[30,31]. BMP1 also modulates the activity and availability of certain growth factors by processing their precursors, thereby influencing cell differentiation and tissue development[32]. Notably, whole-body *Bmp1* knockout mice show

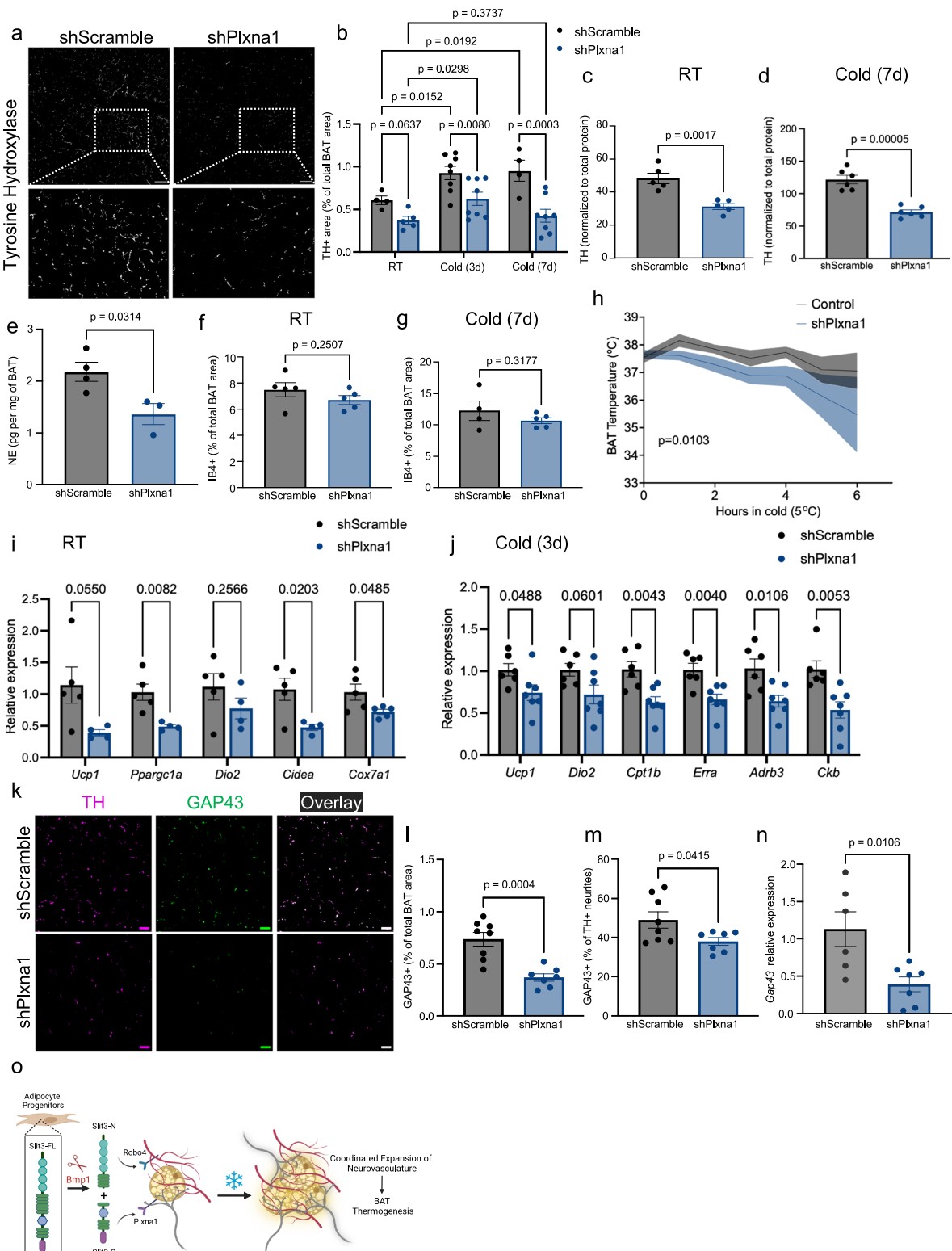

reduced white adipose tissue mass[33]. While it's tempting to speculate that this phenotype may be partly due to impaired SLIT3 cleavage, BMP1's broad substrate specificity makes it difficult to draw direct conclusions from existing data.

Another key finding of our study is the identification of PLXNA1 as the receptor for the C-terminal fragment of SLIT3 and its essential role in regulating sympathetic innervation in BAT. Plexins are a family of single-pass transmembrane proteins that serve as receptors for Semaphorin axon guidance cues. Semaphorin-Plexin signaling

regulates diverse biological processes, including axon guidance, angiogenesis, and immune responses[34]. In vertebrates, class A Plexins typically bind Class 6 Semaphorins and, together with Neuropilin co-receptors, also interact with Class 3 semaphorins[35]. Our data show that PLXNA1 is required for sympathetic innervation of BAT and cold-induced induction of thermogenesis gene programs. Importantly, loss of PLXNA1 eliminates the pro-innervation effects of both SLIT3-C and full-length SLIT3. These findings challenge the conventional view that SLIT proteins exclusively signal through ROBO receptors and that

**Fig. 7 | PLXNA1 is essential for sympathetic innervation and cold-induced neurite expansion in BAT. a** Representative images of TH staining in BAT of mice injected with AAV-shPlxna1 or scramble shRNA. Scale bar = 50 μm. **b** Quantification of TH+ area in BAT from mice housed at room temperature (RT). *n* = 4, 5 animals. Exposed to cold (5 °C) for 3. *n* = 8 animals per group. or 7 days. *n* = 4, 8 animals. **c**, **d** Total TH protein levels in BAT from mice housed at **c** RT. *n* = 5, 5 animals or **d** cold (5 °C) for 7 days. *n* = 6, 6 animals. **e** Norepinephrine concentration normalized to tissue weight in BAT from mice exposed to cold (5 °C) for 3 days. *n* = 4, 3 animals. **f**, **g** Capillary density in BAT from mice housed at **f** RT. *n* = 5, 5 animals, or **g** cold (5 °C) for 7 days. *n* = 4, 5 animals. **h** BAT temperature during the cold tolerance test. *n* = 8 animals per group. **i**, **j** Expression of thermogenic and mitochondrial genes in BAT from mice housed at **i** RT or **j** cold (5 °C) for 3 days.

**k** Representative images of TH and GAP43 staining in BAT from mice housed at cold (5 °C) for 3 days. Scale bar = 20 μm. **l**, **m** Quantification of GAP43+ area (**l**) and percentage of TH+ neurites co-expressing GAP43 (**m**) in BAT from mice housed at 5 °C for 3 days. *n* = 8, 7 animals. **n** Gap43 expression in BAT from mice housed at cold (5 °C) for 3 days. *n* = 6, 7 animals. **o** Proposed model of BAT neurovascular remodeling regulated by Slit3 fragments. Created in BioRender. SHAMSI, F. (2026) https://BioRender.com/qinv3zm. Data are presented as mean ± SEM. Statistical analysis was performed using two-way ANOVA with multiple comparisons correction (Benjamini, Krieger, and Yekutieli) for (**b**); unpaired, two-sided Student's *t* test for (**c**–**g**, **i**, **j**, **l**–**n**); and paired, two-sided Student's *t* test for (**h**). The experiments were repeated two times with similar results. Source data are provided in the Source Data file.

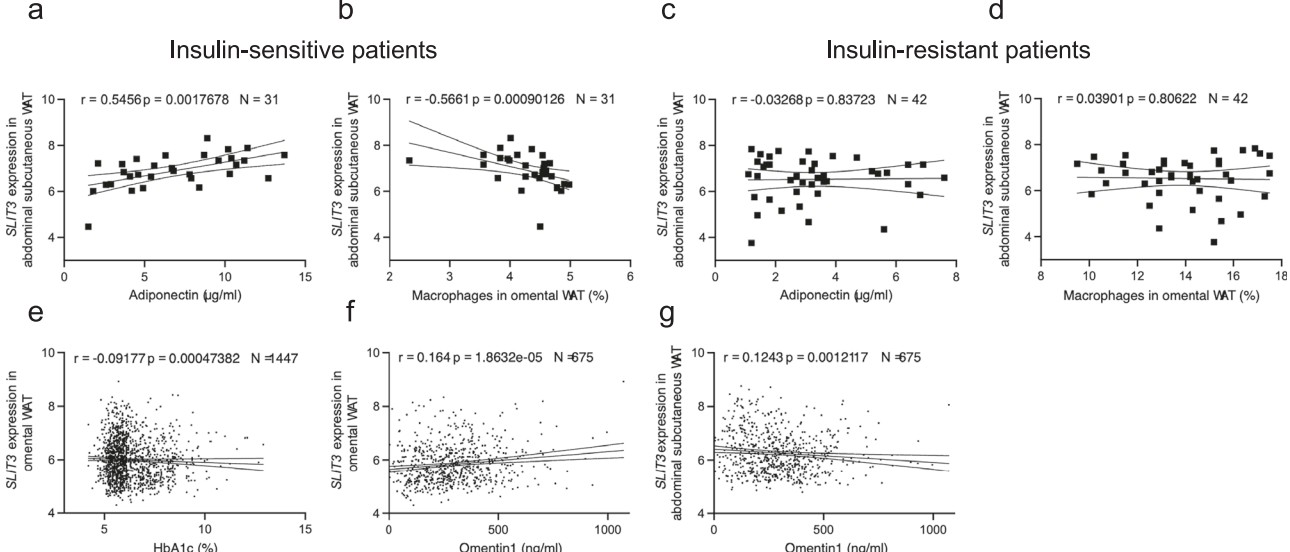

**Fig. 8 | *SLIT3* expression is associated with adipose tissue health and inflammation in humans.** Spearman correlations between SLIT3 expression in adipose tissue and metabolic or inflammatory markers in human cohorts (LOBB). **a**, **b** In insulin-sensitive individuals (MHO/MUO), SLIT3 expression in abdominal subcutaneous WAT correlates with **a** adiponectin (μg/ml) and **b** macrophage content in omental visceral WAT (%). **c**, **d** In insulin-resistant individuals (MHO/MUO), SLIT3 expression in abdominal subcutaneous WAT correlates with **c** adiponectin (μg/ml) and **d** macrophages in omental visceral WAT (%). **e**, **f** In omental visceral WAT (CSC),

SLIT3 expression correlates with **e** HbA1c (%) and **f** Omentin1 levels (ng/ml). **g** SLIT3 expression in abdominal subcutaneous WAT correlates with Omentin1 levels (ng/ml) (CSC). Spearman correlation coefficients (r) associated two-sided *P* values (*P*), and sample size (*N*) are indicated on each plot. Gene expression values represent weighted trimmed mean (TMM) of log expression ratios, with adjustments made for age, sex, and transcript integrity numbers (TINs). Linear regression lines are shown with shaded 95% confidence intervals. Source data are provided in the Source Data file.

Plexins are limited to Semaphorin signaling. Instead, we propose a previously unrecognized level of crosstalk between these two distinct ligand-receptor families. Because PLXNA1 also mediates Semaphorin signaling, we cannot fully exclude the possibility that the reduced innervation observed in PLXNA1-deficient BAT reflects, in part, impaired Semaphorin pathways. Future studies will be important to dissect the relative contributions of SLIT3 and Semaphorin signaling through PLXNA1 in regulating sympathetic innervation.

Adipose tissues are extensively innervated by a network of sympathetic and sensory nerve projections, which facilitate the transmission of information between the adipose tissue and the central nervous system. It is well-established that the sympathetic innervation of adipose tissue is critical for homeostatic control of its function and whole-body metabolism[9,36,37]. The plasticity of sympathetic nerves allows for altered neuronal control and adaptation to metabolic challenges. However, the mechanisms underlying the local expansion and remodeling of sympathetic neurites are not well understood. Obesity disrupts norepinephrine-mediated adipocyte lipolysis[38], mitochondrial biogenesis[39], and adipose tissue remodeling[40]. These impairments are attributed to dysfunctional local sympathetic innervation resulting from adipose inflammation[41]. In turn, the impaired sympathetic

activity in obese adipose tissue may contribute to more inflammation and further exacerbate adipose dysfunction[42]. Therefore, preserving the density and functionality of sympathetic nerves may help safeguard adipose tissue health in obesity. To the best of our knowledge, this study is the first to demonstrate that adipocyte progenitors play a regulatory role in sympathetic innervation of adipose tissue depots, introducing a previously unexplored regulator of adipose tissue innervation.

Genome-wide association studies have found genetic variants near the *SLIT3* gene to be associated with an increased risk of insulin resistance[43] and higher BMI[44]. Additionally, a differentially methylated CpG in the *SLIT3* gene in visceral adipose tissue was found to be associated with the development of type 2 diabetes in obese women[45]. Genetic variants in the *PLXNA1* gene are also associated with severe early onset obesity[46]. These independent findings suggest that SLIT3 and its receptor PLXNA1 may have a role in regulating metabolism in humans. Our analysis of human adipose tissue in two obesity cohorts suggest that SLIT3 signaling may contribute to improved adipose tissue health, reduced inflammation, and insulin sensitivity in human obesity. These findings highlight the need for further investigation into SLIT3-PLXNA1 signaling in the context of obesity and metabolic disease.

Limitations of the study: While our findings demonstrate that *Bmp1* knockdown or inhibition blunts SLIT3 proteolysis, definitive confirmation will require direct biochemical characterization of BMP1 activity on SLIT3, including substrate specificity and kinetic analyses. Similarly, although our structural modeling, biochemical assays, and in vivo data strongly support PLXNA1 as a receptor mediating SLIT3-C signaling, quantitative binding analyses will be needed to delineate differences in receptor binding affinity and specificity between SLIT3-FL and its proteolytic fragments. In addition to *Pdgfra*⁺ APCs, scRNA-seq and analyses of sorted primary cells revealed *Slit3* expression in adipose endothelial cells. Future studies employing endothelial cell-specific loss-of-function models will be important to define the contribution of endothelial-derived SLIT3 to adipose tissue function and remodeling. Given the established role of ROBO4-mediated signaling in angiogenesis, we did not directly test the necessity of endothelial ROBO4 for SLIT3-N-induced angiogenesis in BAT. Finally, the observed associations between *SLIT3* expression and markers of adipose tissue health, such as increased adiponectin levels and reduced macrophage abundance, provide evidence linking the SLIT3 pathway to human adipose tissue function. However, these findings are correlative and do not establish causality. Further investigations are warranted to determine the physiological significance of Slit3 signaling in human adipose tissue and metabolic health.

## Methods

### Animals
All experiments were performed in compliance with all relevant ethical regulations and were approved by Research and Laboratory Safety and the Institutional Animal Care and Use Committees at New York University.

C57BL/6 J mice (stock no. 000664) were purchased from Jackson Laboratory. Pdgfra-cre;Slit3^flox/flox or Slit3^iΔAPC were generated by crossing Slit3 floxed mice[20] with Pdgfra-creER^TM[21] (JAX strains 018280). Mice were maintained on a 12-h light-dark cycle at 22 °C and 50% relative humidity. Mice were maintained on a regular chow diet (5053 - PicoLab® Rodent Diet 20; formulated with 20% protein diet and 4.5% fat) with food and water provided ad libitum unless otherwise specified. For experiments involving cold acclimation or thermoneutral housing, mice were maintained at 5 °C and 30 °C, respectively, with 50–60% relative humidity in a controlled diurnal environmental chamber (Caron Products & Services) and given free access to food and water. Euthanasia of the animals was accomplished by methods consistent with the guidelines of the American Veterinary Medical Association. Euthanasia was performed by $CO_2$ inhalation in accordance with institutional guidelines, followed by a secondary method (cervical dislocation) to ensure death.

### Cold tolerance test
For cold tolerance tests, mice were single-housed and placed at 5 °C in a controlled environmental diurnal chamber (Caron Products & Services). Body temperature was measured with a rectal probe (Physitemp, RET3) and a reader (Physitemp, BAT-12) or using RFID transponder temperature microchips implanted under the interscapular BAT area (Unified Information Devices).

### AAV injection
6–8-week-old male mice were anesthetized with isoflurane, and an incision was made above the interscapular area to expose the underlying adipose tissue. AAV particles were injected into each BAT lobe, and the incision was closed with suture. Mice were allowed to recover for 2–3 weeks before analysis.

### Slit3 and Plxna1 shRNAs
Adeno-associated viruses (AAV8) harboring three targeting shRNA and one scramble control were purchased from VectorBuilder. The shRNA sequences are listed in Supplementary Table 1. The three shRNAs were mixed and injected into BAT at the dose of 5e + 11 genomic copies per BAT lobe.

### AAV constructs for the overexpression of Slit3 fragments in BAT
Mouse SLIT3-FL, SLIT3-N, and SLIT3-C sequences were cloned from a cDNA clone (Origene MR225499L4) and cloned into the pAAV-hAdiponectin-W backbone[47] using NEBuilder HiFi DNA Assembly kit using the following primers:

Slit3-FL_fwd (AACTACTCGAGgccaccATGGCCCTCGGCCGGAC),
Slit3-FL_rev (tattcaGCGGCCGCTTAAACCTTATCGTCGTCATCCTT GTAATCgctgccGGAACACGCGCGGCA), Slit3-N_fwd (AACTACTCGAG gccaccATGGCCCTCGGCCGGAC), Slit3-N_rev (tattcaGCGGCCGCT-TAAACCTTATCGTCGTCATCCTTGTAATCgctgccAACCATGGGTGGGG G), Slit3-C_fwd (tgctgccgccCTGCTACAAACCAGCCCC), Slit3-C_rev (ggttgattatcttctagagcTTACTTGTCGTCATCGTCTTTG), Signal pepti-de_fwd (tgattccataccagagggtcGCCACCATGGCCCTCGGC), and Signal peptide_rev (tttgtagcagGGCGGCAGCAGGGGGTCC).

pAAV-Adipoq-GFP construct[47] was used as the control. AAV8-Adipoq-Slit3FL, AAV8-adipoq-Slit3N, AAV8-Adipoq-Slit3C, and AAV-Adipoq-GFP were packaged at VectorBuilder. 2e + 10 AAV particles were injected into each BAT lobe.

### Fluorescence-activated cell sorting (FACS)
Adipocytes and the Stromal Vascular Fraction (SVF) were isolated from mouse BAT following the procedure described previously[5]. The interscapular BAT was dissected, finely minced, and digested for 45 min using a cocktail containing type 1 collagenase (1.5 mg ml⁻¹; Worthington Biochemical), dispase II (2.5 U ml⁻¹; Stemcell Technologies), and fatty acid-free BSA (2%; Gemini Bio-Products) in Hanks' balanced salt solution (Corning Hanks' Balanced Salt Solution, with calcium and magnesium). The resulting dissociated tissue was subsequently centrifuged at 500× g and 4 °C for 10 min. Adipocytes, located in the uppermost layer, were gently collected using a wide-mouthed transfer pipette and filtered through a 100 μm cell strainer. Brown adipocytes were allowed to float for 5 min at room temperature before they were centrifuged at 30 g for 5 min at room temperature. This cycle was repeated three times, after which the adipocytes were immediately lysed in Trizol. For the SVF isolation, the pellet was resuspended in 10 ml of 10% FBS in DMEM, filtered through a 100 μm cell strainer into a fresh 50-ml tube, and subsequently centrifuged at 500× g for 7 min. Red blood cells were lysed in 2 ml of sterile ammonium-chloride-potassium lysis buffer (ACK Lysis Buffer, Lonza) for 5 min on ice. The cells were then filtered once more through a 40-μm cell strainer, washed with 20 ml of a solution containing 10% FBS in DMEM, and centrifuged at 500× g for 7 min. The cells were resuspended in 1 ml of Cell Staining Buffer (BioLegend) before proceeding with staining.

Cells were stained using the fluorescently conjugated antibodies as outlined in Supplementary Table 2. The cells were then incubated with the antibodies at the specified dilutions from Supplementary Table 2 for a duration of 30 min, followed by two rounds of washing in Cell Staining Buffer (BioLegend). Cells were sorted using an SH800 sorter using a 100 μm sorting chip (Sony Biotechnology). Debris and doublets were excluded based on forward and side scatter gating, and 7-AAD was used to exclude dead cells. Following sorting, cells were centrifuged at 300× g for 5 min and lysed in Trizol for subsequent RNA isolation and gene expression analysis.

### RNA isolation and quantitative reverse transcription-PCR (qRT-PCR)
RNA was isolated from cells or tissues using phenol–chloroform extraction and isopropanol precipitation. qRT PCR assays were conducted using a QuantStudio™ 5 Real-Time PCR instrument (Life Technologies Corporation) and SYBR Green (Invitrogen). Relative mRNA expression was determined using the ΔCt method and

normalized to the expression of housekeeping genes Rplp0 or Tbp. Primer sequences are available in Supplementary Table 3.

## Western blotting

Cells or tissues were lysed in RIPA buffer supplemented with a protease inhibitor cocktail (cOmplete™, Sigma-Aldrich, Dallas, TX). The primary antibodies are listed in Supplementary Table 2. Primary antibodies were incubated overnight at 4 °C. HRP-coupled secondary antibodies were used at 1:5000 dilution for 1 hour at room temperature. Proteins were detected using the Pierce™ ECL Western Blotting Substrate (ThermoFisher Scientific) using an Licor Odyssey M Imaging System. All the original uncropped and unprocessed scans are provided in the Source Data file. For protein detection in the conditioned media, serum-free and Phenol Red-free media were collected and centrifuged at $300 \times g$ to remove cell debris. The supernatant was concentrated using Amicon™ Ultra-4 Centrifugal Filter Units (UFC801024, MilliporeSigma™).

## Immunohistochemistry

Adipose tissues were fixed in 10% formalin and embedded in paraffin. 6 μm sections were prepared and stained with the primary antibodies listed in Supplementary Table 2. Sections were deparaffinized and rehydrated, followed by an antigen retrieval step in Rodent Decloaker buffer using the Decloaking Chamber™ NxGen (BIOC Medical) in 1× rodent antigen retrieval reagent (95 °C for 45 min). Sections were then incubated in Sudan Black (0.3% in 70% ethanol) to reduce the auto-fluorescence signal. Blocking was performed in 5% BSA in PBST (0.1% Tween-20 in PBS) for 30 minutes at room temperature, followed by incubating the sections in primary antibodies diluted in 1% BSA in PBST overnight at 4 °C (Supplementary Table 2). The biotinylated isolectin B4 was diluted in 5 μg/ml in a staining buffer containing 0.1 mM CaCl₂, 0.1 mM MgCl₂, and 0.1 mM MnCl₂. The next day, slides were washed with PBST and were incubated with appropriate fluorescently labeled secondary antibodies at a 1:200 dilution (Supplementary Table 2). Slides were stained with DAPI and mounted in a mounting medium. Images were collected on a Leica SP8 confocal microscope and processed with ImageJ.

## H&E staining

Adipose tissues were fixed in 10% formalin and embedded in paraffin. 6 μm sections were prepared and stained with hematoxylin and eosin (H&E) as previously described[48]. Slides were imaged using an Aperio slide scanner (Leica Biosystems).

## Indirect calorimetry

Mice were individually housed in metabolic cages of a TSE Phenomaster system to obtain measurements for the volume of oxygen consumption (VO2), the volume of carbon dioxide production (VCO2), respiratory exchange ratio (RER), energy expenditure, food intake, and locomotor activity. Mice had ad libitum access to drinking water and standard rodent chow. Measurements were collected first for 48 hour at room temperature, followed by the second 48 hour at 5 °C. Data were analyzed using CalR[49].

## Dual energy X-ray absorptiometry

Lean and fat mass were quantified using a Dual Energy X-ray Absorptiometry (DEXA) scanner (Insight, Osteosys).

## Tissue clearing and imaging

Whole BAT sympathetic innervation visualization was achieved using the Adipo-Clear protocol[50] with some modifications. One lobe of BAT from each animal was fixed in 3% glyoxal at 4 °C overnight. The fixed samples were washed in PBS three times, each time for 1 hour. Next, the samples were subjected to a series of methanol washes: 20%, 40%, 60%, 80% methanol in H2O/0.1% Triton X-100/0.3 M glycine (B1N

buffer, pH 7), and finally 100% methanol, each for 30 minutes. Subsequently, the samples underwent a triple 30-minute delipidation process with 100% dichloromethane (DCM; Sigma-Aldrich) followed by two 30-minute washes in 100% methanol. After delipidation, the tissues were bleached overnight at 4 °C with 5% H₂O₂ in methanol (1 volume of 30% H₂O₂ to 5 volumes of methanol). The rehydration process followed a reversed methanol/B1N buffer series: 80%, 60%, 40%, 20% methanol in B1N buffer, each step lasting 30 minutes. All the above steps were conducted at 4 °C with continuous shaking. Subsequently, samples were washed in B1N buffer twice, each time for 30 minutes, followed by two 1-hour wash in PBS/0.1% Triton X-100/ 0.05% Tween-20/2 μg/ml heparin (PTwH buffer), before initiating the staining procedure. The samples were incubated with anti-tyrosine hydroxylase antibody (1:200, AB1542, Millipore Sigma) in PTxwH for 4 days. After the primary antibody incubation, the samples underwent a series of PTxwH washes lasting 5 minutes, 10 minutes, 15 minutes, 30 minutes, 1 hour, 2 hours, 4 hours, and overnight, and then were incubated in donkey anti-sheep IgG Alexa Fluor 647 (1:200, A-21099, Thermo Fisher) in PTxwH for 4 days. Finally, the samples were washed in PTwH for 5 minutes, 10 minutes, 15 minutes, 30 minutes, 1 hour, 2 hour, 4 hour, and overnight. Finally, the samples were dehydrated using a methanol/H₂O series (25%, 50%, 75%, 100%, 100%) for 30 minutes at each step at room temperature. Following dehydration, samples were incubated with 100% DCM for 30 minutes twice, followed by an overnight clearing step in dibenzyl ether (DBE; Sigma-Aldrich). The samples were then stored at room temperature in the dark until imaging. Representative samples were imaged by a blinded experimenter using a Zeiss Lightsheet Z.1 Microscope with two 1920 × 1920-pixel sCMOS cameras. Three-dimensional reconstructions were generated using Imaris (Bitplane).

## Cloning of SNAP- and Halo-tagged Slit3 plasmids

pRP[Exp]-CMV > [SNAP-Slit3-HaloTag], pRP[Exp]-CMV > [Slit3-FL]-HaloTag, and pRP[Exp]-CMV > [Slit3-C-HaloTag] constructs were generated by VectorBuilder. The N-terminal tag was added immediately downstream of the signal peptide (aa 2-33, GCCCTCGGCCGGA CCGGGGCCGGCGCCGCTGTGCGCGCCCGCCTGGCGCTGGGCTTGGC-GCTTGCGAGCATCCTGAGCGGACCCCCTGCTGCCGCC).

The ORF in the pRP[Exp]-CMV > [SNAP-Slit3UC-Halo] was generated by the deletion of a 27 bp sequence encoding the Slit cleavage site (CCCACCCATGGTTCTGCTACAAACCAG, PPPMVLLQ) from Slit3-FL cDNA.

## Transient transfection

Immortalized brown preadipocytes were transfected with Slit3 over-expression plasmids, Bmp1 or scramble siRNA (Dharmacon GE) using Lipofectamine™ 3000 or Lipofectamine™ RNAiMAX Transfection Reagents (Invitrogen™) following the manufacturer's instructions.

## Bmp1 inhibitor treatment

Cells were treated with the Bmp1 inhibitor, UK-383,367 (PZ0156-5MG, Sigma Aldrich) at the final concentration of 2.5 μM.

## AlphaFold multimer analysis

AlphaFold2 Multimer was used to predict the protein-protein interactions between the extracellular domains of human PLXNA1 and the C-terminal region of the Slit3 protein, as described previously[51,52]. Protein sequences used in this study are as follows: human PLXNA1 residues 27-1244 (UniProt ID: Q9UIW2), human Slit3 residues 1120-1523 (UniProt ID: O75094), human Semaphorin-6D residues 1-1073 (UniProt ID: Q8NFY4). Briefly, a local graphics processing unit (GPU) cluster using MMseqs (git@92deb92) was utilized for local Multiple Sequence Alignment (MSA) creation[53], and ColabFold (git@7227d4c) was executed for structure prediction with 5 models per prediction, including structure relaxation[54]. Predictions with an average pTM+ iPTM score of

Article

>0.8 were ranked, and diagnostic plots (PAE plot, pLDDT plot, and sequence coverage) were manually inspected. The generated three-dimensional (3D) models of protein-protein complexes were analyzed using UCSF Chimera[55] and the PyMOL Molecular Graphics System, Version 3.0 Schrödinger, LLC.

## Norepinephrine measurement

BAT samples were lysed in buffer containing 40 mM sodium metabisulfite and 10 mM EDTA. Homogenates were centrifuged at 12,000 ×$g$ for 5 min at 4 °C, and the supernatants were collected for analysis. Norepinephrine levels were quantified using an ELISA kit (Biomatik, EKF577994-96T) according to the manufacturer's instructions.

## Human data

The human data used in this research were sourced from the Leipzig Obesity Biobank (LOBB, https://www.helmholtz-munich.de/en/hi-mag/cohort/leipzig-obesity-bio-bank-lobb), which comprises paired samples of abdominal subcutaneous and omental visceral adipose tissue. The metabolically healthy versus unhealthy obese cohort (MHO/MUO) comprises paired samples of omental visceral and abdominal subcutaneous tissues from 31 insulin-sensitive patients (71% women; age: 38.8 ± 11.1 years old; BMI: 45.9 ± 6.9 kg/m²; fasting plasma glucose: 5.2 ± 0.2 mmol/l; fasting plasma insulin: 27.9 ± 13.5 pmol/l) and 42 insulin-resistant patients (71.43% female; age: 47.2 ± 7.7 years old; BMI: 47.3 ± 8.1 kg/m²; fasting plasma glucose: 5.7 ± 0.3 mmol/l; fasting plasma insulin: 113.7 ± 45.7 pmol/l). The cross-sectional cohort (CSC) comprises 1,479 individuals, categorized as either normal/overweight (N = 31; 52% women; age: 55.8 ± 13.4 years; BMI: 25.7 ± 2.7 kg/m²) or obese (N = 1448; 71% women; age: 46.9 ± 11.7 years; BMI: 49.2 ± 8.3 kg/m²). Adipose tissue samples were collected during elective laparoscopic abdominal surgeries, following established protocols[56,57]. Body composition and metabolic parameters were assessed using standardized techniques as described previously[58,59]. The study was approved by the Ethics Committee of the University of Leipzig (approval numbers: 363-10-13122010 and 017-12-230112) and adhered to the principles outlined in the Declaration of Helsinki. All participants provided written informed consent before being included in the study. Participants received no compensation for their tissue donation to the Leipzig Obesity BioBank. Exclusion criteria included individuals under 18 years of age, those with chronic substance or alcohol abuse, smoking within the 12 months prior to surgery, acute inflammatory conditions, concurrent use of glitazones, end-stage malignancies, weight loss greater than 3% in the three months leading up to surgery, uncontrolled thyroid disorders, and Cushing's disease.

Ribosomal RNA-depleted RNA sequencing data we generated following the SMARTseq protocol[60]. The libraries were sequenced as single-end reads on a Novaseq 6000 (Illumina, San Diego, CA, USA) at the Functional Genomics Center Zurich, Switzerland. The pre-processing procedures were conducted as previously described[61]. In brief, adapter and quality-trimmed reads were aligned to the human reference genome (assembly GRCh38.p13, GENCODE release 32), and gene-level expression quantification was performed using Kallisto[62] v0.48. For samples with read counts exceeding 20 million, we downsampled them to 20 million reads utilizing the R package ezRun v3.14.1 (https://github.com/uzh/ezRun, accessed on April 27, 2023). The data normalization was carried out using a weighted trimmed mean (TMM) of the log expression ratios, with adjustments made for age, sex, and transcript integrity numbers (TINs). Analyses were conducted in R v4.3.1 (www.R-project.org). Spearman coefficient was used to assess the correlation between *SLIT3* expression and metabolic parameters.

## Statistics and reproducibility

No statistical method was used to predetermine sample size. No data were excluded from the analyses. Experiments were not randomized.

All imaging and quantifications were performed in a blinded manner. Statistical analyses were performed using GraphPad Prism.

## Reporting summary

Further information on research design is available in the Nature Portfolio Reporting Summary linked to this article.

## Data availability

The data supporting the findings of this study are available within the paper and its supplementary information files. Proteomic dataset used in Fig. 4d has been deposited to Proteomexchange PRIDE under accession PXD035318. The scRNA-seq data used in this study are available in the GEO database under accession codes GSE160585 and GSE176067 and in the SRA database under accession code SRP322736. Access to the human RNA-seq data from the LOBB study is regulated by the LOBB steering committee and requires an approved Data Use Agreement (DUA) outlining the permitted research purposes and protections for participant privacy. Requests will be acknowledged within 5–10 business days and, following review, granted or denied typically within 2–4 weeks, depending on the completeness of the application and the scope of the requested data. Access is restricted to non-commercial, non-identifiable research use. Data access may be granted to named researchers or institutions after execution of a DUA and approval by the steering committee. For access, contact Matthias Blüher (matthias.blueher@medizin.uni-leipzig.de) or Anne Hoffmann (anne.hoffmann@helmholtz-munich.de). Source data are provided with this paper.

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

## Acknowledgements

This work was supported in part by the US National Institutes of Health (grants K01DK125608, R03DK135786, R01DK136724), an award from The G. Harold and Leila Y. Mathers Charitable Foundation, the American Heart Association Career Development Award (24CDA1271852), a grant from Einstein-Mount Sinai Diabetes Center, and the Department of Molecular Pathobiology Accelerator Award (to F.S.), by National Institutes of Health grant RC2DK129961 (to P.C.), R35GM150942 (to H.A.), and Boettcher Foundation Webb-Waring Biomedical Research Award (to H.A.). The Albert Einstein Animal Physiology Core is supported by the NIH grant DK20541. M.B. received funding from grants from the DFG project number 209933838 – SFB 1052 (project B1) and by Deutsches Zentrum für Diabetesforschung (DZD, Grant: 82DZD00601). We thank Adam Mar and Begona Gamallo-Lana at the NYU Langone Rodent Behavior Laboratory (RRID: SCR_017942) for providing technical assistance, Experimental Pathology Research Laboratory (RRID:SCR_017928), Preclinical Imaging Laboratory (RRID:SCR_017937), and Microscopy Laboratory (RRID: SCR_017934). The microscopy shared resource is partially supported by Cancer Center Support Grant P30CA016087.

## Author contributions

T.D.S. and F.S. conceptualized the work and designed research. T.D.S., B.F., H.C., Q.T., and D.H. performed research and analyzed data. A.G.I. and H.A. performed the AlphaFold Multimer Analysis. C.H.J.C. and P.C. provided the adipocyte secretome data. A.H. performed RNA sequencing and preprocessing of the LOBB data. M.B., A.G., and C.W. provided the data from the LOBB cohorts. M.G. provided the Slit3 flox mouse strain. G.J.S. contributed to experimental design and provided feedback on data interpretation and manuscript preparation. F.S. wrote the manuscript with input from all authors.

## Competing interests

F.S. is a scientific co-founder and serves on the advisory board of Noara Therapeutics. M.B. received honoraria as a consultant and speaker from Amgen, AstraZeneca, Bayer, Boehringer-Ingelheim, Lilly, Novo Nordisk, Novartis, and Sanofi. The remaining authors declare no competing interests.

## Additional information

[1]Department of Molecular Pathobiology, College of Dentistry, New York University, New York, NY, USA. [2]Laboratory of Molecular Metabolism, The Rockefeller University, New York, NY, USA. [3]Helmholtz Institute for Metabolic, Obesity and Vascular Research (HI-MAG) of the Helmholtz Center Munich at the University of Leipzig and University Hospital Leipzig, Leipzig, Germany. [4]Institute of Food, Nutrition and Health, ETH Zurich, Schwerzenbach, Switzerland. [5]Department of Pathology and Laboratory Medicine, Weill Cornell Medical College, New York, NY, USA. [6]Medical Department III – Endocrinology, Nephrology, Rheumatology, University of Leipzig Medical Center, Leipzig, Germany. [7]Pain Research Center, New York University, New York, NY, USA. [8]Department of Biochemistry and Molecular Pharmacology, Grossman School of Medicine, New York University, New York, NY, USA. [9]Department of Medicine, Fleischer Institute for Diabetes and Metabolism, Albert Einstein College of Medicine, Bronx, NY, USA. [10]Department of Neuroscience, Fleischer Institute for Diabetes and Metabolism, Albert Einstein College of Medicine, Bronx, NY, USA. [11]Department of Cell Biology, Grossman School of Medicine, New York University, New York, NY, USA. [12]Department of Medicine, Grossman School of Medicine, New York University, New York, NY, USA. [13]These authors contributed equally: Heidi Cervantes, Benjamin Frank. ✉e-mail: fs2451@nyu.edu

