## [Transparent Peer Review file · Nature Communications]

SLIT3 Fragments Orchestrate Neurovascular Expansion and Thermogenesis in Brown Adipose Tissue

Corresponding Author: Dr Farnaz Shamsi

Version 0:

Reviewer comments:

Reviewer #1

(Remarks to the Author)

In this manuscript, the authors identify Slit3 as a key factor mediating the neurovascular organization of brown adipose tissue (BAT) in mice. Using single-cell analyses and western blotting, they confirm that Slit3 is expressed in adipose tissue progenitor cells. They further demonstrate that Slit3 enhances vascularization and innervation of BAT in response to cold exposure and plays a major role in supporting thermogenesis under these conditions. The functional effects are attributed to the C-terminal fragment of Slit3, processed by BMP1 and signaling through Plxna1. Overall, the experiments are well described and the results largely support the authors' conclusions. However, certain aspects require additional evidence or clarification.

Specific Points:

The authors report that reduction of Slit3 leads to whitening of BAT under cold conditions but do not provide an explanation for this observation. Is BAT accumulating lipids released from lipolysis in other tissues? What are the weights of the inguinal and epididymal fat pads in these mice?

Angiogenesis is assessed via isolectin staining. However, because adipocytes enlarge under certain conditions, vessel density per field can appear reduced. This must be controlled for by normalizing to adipocyte density—for example, expressing isolectin intensity relative to LipiTox or PLIN1 staining intensity. The same normalization should apply to TH staining.

In shRNA-treated mice, BAT shows a marked increase in lipid droplet size. Is this phenotype recapitulated in the Pdgfra-Cre-driven Slit3 knockout (Fig. 3)? If so, isolectin and TH staining data must also be corrected for adipocyte density, as recommended for Fig. 2.

Enforced AAV-mediated expression of the C-terminal or full-length Slit3 increases isolectin staining, suggesting enhanced angiogenesis (Fig. 5). However, adipocyte density should be reported, and the data normalized accordingly. The rationale for performing these experiments under conditions where endogenous Slit3 expression is already expected to be high (WT mice after 7 days of cold exposure) is unclear. A more compelling experiment would assess the effect of Slit3-C in non-cold-exposed mice, where any increase in innervation could be attributed directly to the construct.

The hypothesis that BMP1 functions as a protease for Slit3 is supported by data showing diminished proteolysis following siRNA-mediated Bmp1 knockdown or pharmacological inhibition. While these findings suggest a requirement for BMP1 in Slit3 cleavage, they do not definitively establish BMP1 as the responsible protease. Rigorous molecular characterization—including specificity and kinetic studies—would be required to confirm this. Similarly, while structural modeling and shRNA knockdown data support the role of Plxna1 as a receptor for Slit3-C, definitive evidence requires demonstration of direct receptor-ligand binding, including affinity and specificity measurements. Although these experiments may be beyond the scope of the current study, the limitations of the conclusions should be clearly acknowledged in the manuscript.

Reviewer #2

(Remarks to the Author)

General Assessment:

This study investigates the molecular mechanisms coordinating neurovascular expansion and thermogenesis in brown adipose tissue (BAT), with a focus on the role of adipocyte progenitors. Building on previous scRNA-seq findings that identified adipocyte progenitors as key communication hubs, the authors perform in-depth bioinformatics analyses to identify secreted ligands mediating intercellular crosstalk. They identify secreted axon guidance molecule Slit3 and demonstrate through functional experiments that Slit3 is essential for BAT thermogenesis, angiogenesis, and sympathetic innervation.

The study further reveals that Slit3 is proteolytically processed by Bmp1 into distinct fragments (Slit3-FL, Slit3-N, and Slit3-C), each with specific roles in neurovascular remodeling. AAV-mediated gain-of-function experiments show that Slit3-C promotes sympathetic innervation, while Slit3-N enhances angiogenesis. The authors also identify Plxna1 as a receptor mediating Slit3-C-dependent sympathetic innervation. Finally, the study links Slit3 expression to adipose tissue health and inflammation in humans.

This is a well-structured and compelling study that provides novel mechanistic insights into how adipocyte progenitors orchestrate neurovascular remodeling in BAT. The findings are timely and significantly advance our understanding of BAT biology, and the experimental design is generally robust. While there are no major concerns, the following suggestions may help further strengthen the manuscript:

Specific Comments:

1. Figure 1C: The increase in Slit3-FL under cold conditions is shown, but it is unclear whether similar changes occur for Slit3-N or Slit3-C. If the antibody used is expected to recognize Slit3-N or Slit3-C, this should be clarified. Including this information as supplementary data would enhance transparency.
2. Supplementary Figure 2a–d: To support the claim that Slit3 is not expressed in macrophages under basal or cold conditions, immunofluorescence staining could provide additional confirmation.
3. Supplementary Figure 2e: The use of a mouse adipocyte progenitor cell line to assess Slit3 expression during adipogenesis is informative. However, staining for Slit3 in both mature adipocytes and progenitors in BAT and WAT would provide more comprehensive validation.
4. Figure 2g: A protein loading control is missing. Including this would strengthen the interpretation of the immunoblot data.
5. Figure 3a: Similarly, a loading control is needed. Given that Slit3 is expressed in multiple cell types in BAT, and that Slit3 expression is expected to be reduced but not absent in Slit3iDAPC tissue, it would be helpful to show Slit3 protein levels under both basal and cold conditions with appropriate controls.
6. Figure 5g–h: Overexpression of Slit3-C also increases capillary density in BAT. This is unexpected given the fragment's primary role in sympathetic innervation. A brief discussion of potential mechanisms would be valuable.
7. Figure 6d: The lack of co-immunoprecipitation between Plxna1 and Slit3-N supports the specificity of Plxna1 for Slit3-C. This point could be emphasized more clearly to strengthen the conclusion.
8. Figure 1i: The p-value for Dio2 is 0.062. While not statistically significant, the trend may still be biologically relevant and could be briefly discussed.
9. Line 291: There appears to be a typo—"Slit3-FL" should be "Slit3-C."
10. Supplementary Figure 6e: The high-magnification images do not appear to align with the low-magnification views. Please verify and adjust if necessary.
11. Literature Context: Including a brief section summarizing the Slit-Robo signaling pathway and its known roles in neurogenesis and angiogenesis in the Introduction would provide helpful context for readers less familiar with this signaling axis.

Reviewer #3

(Remarks to the Author)

Cold exposure leads to a marked increase in vascularization and sympathetic nervous system (SNS) innervation of thermogenic adipose tissue to meet increased energetic demands. Several independent groups implicated neurotrophins and Slit3 secreted by myeloid cells in the cold-induced expansion of sympathetic innervation in inguinal white adipose tissue (iWAT) (PMID: 33297933, PMID: 35042776; PMID: 29589323; PMID: 34782792). This manuscript identifies key molecular mediators of cold-induced neurovascular expansion in brown adipose tissue (BAT). First, the authors demonstrate that Slit3 is predominantly expressed in adipocyte progenitor cells (APCs) and vascular smooth muscle in BAT, and not myeloid cells as in iWAT. Complementary loss-of-function studies using AAV-mediated expression of shRNAs in BAT and genetic knockout in APCs provide compelling evidence that Slit3 is necessary for cold-induced increases in angiogenesis and sympathetic innervation in BAT. In vitro loss-of-function and in vivo gain of function studies support the idea that the

metalloprotease BMP1 cleaves Slit3 into two fragments that regulate distinct functions. The N-terminal fragment promotes angiogenesis (consistent with PMID: 22641771), while the C-terminal fragment increases SNS innervation. The authors identified Plxn1 as a potential receptor for the C-terminal fragment of Slit3 using computational modeling. Complementary in vitro and in vivo studies support the idea that Plxn1 is a key receptor mediating the effects of the full length or C-terminal fragment of Slit3 on SNS innervation in BAT. Expression studies identified Robo4 as a putative receptor on endothelial cells, but its requirement for cold-induced angiogenesis was not tested.

In summary, while Slit3 appears to regulate similar adaptive responses to cold in iWAT and BAT, these studies show that the source of Slit3 (myeloid cells vs APCs) and the specific receptors mediating effects on the SNS and vasculature are distinct (Robo1 vs Plxn1). The novelty of these studies lies in the demonstration that BMP1 cleaves Slit3 into 2 fragments that act on the SNS and vasculature via distinct receptors. The authors propose that the distinct activities of Slit3 proteolytic cleavage products provide a means to synchronize adaptive responses to cold. Overall, the complementary and rigorous approaches provide very strong support for the molecular pathways outlined here. However, there are several outstanding issues that need to be resolved, outlined below.

1 The most fundamental outstanding issue is whether Slit3 only functions in the context of cold-induced remodeling or also maintains the neurovascular architecture at thermoneutrality (i.e. baseline). This is critical to know if you want to translate these findings to humans, who live near thermoneutrality.

Slit3 KD reduced neurovascular density at room temperature (Suppl Fig 7), which is a mild cold challenge. Analyzing Slit3 KD or cKO models at thermoneutrality would be informative as to whether Slit3's function is restricted to cold-induced expansion or also affects baseline structure.

2. Whereas APCs express the highest levels of Slit3, cold-induced increases in transcription are only observed in ECs (Suppl Fig 1d). Moreover, AAV-mediated Slit3 KD in BAT (Fig. 1e) produced a stronger reduction in body temperature after 5 hr at 5C than conditional KO in Pdgfr+ APCs (Figure 3d) (KD = 33.5C vs KO= 36.5C). This raises the possibility that adipocyte-derived Slit3 helps to maintain the neurovascular architecture at baseline but that EC-derived Slit3 mediates cold-induced remodeling. Comparisons between adipocyte- and EC-specific Slit3 KOs at thermoneutrality and in response to a cold challenge would resolve this issue as well as issue #1.

3. While the data supporting a role for Slit3 in cold-induced neurovascular remodeling are compelling, the relationship with BAT thermogenesis is less clear. In the Slit3 KD, BAT temperature (Suppl Fig 3e) and body temperature (Fig 1e) are lower in the first 5 hours of cold exposure – this is before any cold-induced remodeling occurs. On the other hand, after 2 days of cold exposure, when Slit3 levels are high (Figure 1d), BAT temperature of KDs is the same as controls (Suppl Fig 3e). Additionally, Plxn1 KD in conjunction with Slit3-C overexpression prevents the cold-induced increase in SNS innervation (i.e. innervation looks like controls), but the BAT temperature curve is similar to that of the Plxn1 KD alone, which has lower SNS innervation (Figure 6 l,m).

Unless these apparent inconsistencies can be definitively resolved, I would recommend removing the body/BAT temperature data and narrowing the claims to SNS and vascular remodeling.

4. Analyses of mice focused on the impacts of the transition from a moderate (22C) to a severe (5C) cold challenge on endpoints related to neurovascular remodeling in lean animals. In contrast, samples of abdominal subcutaneous and visceral omental WAT depots were obtained from obese people living near thermoneutrality. The most compelling associations are a positive correlation between Slit3 and adiponectin expression in abdominal subcutaneous WAT and a negative correlation with macrophages in omental visceral WAT (which could be due to the increase in adiponectin). Analysis of markers of SNS tone (i.e. TH) or capillaries would help to connect Figure 8 to the rest of the manuscript. Otherwise, the manuscript would be strengthened by saving these interesting human data for another manuscript.

Minor Concerns:

1. The authors posit that “the co-regulation of neurovascular expansion by distinct Slit3 fragments offers a bifurcated yet harmonized mechanism to ensure a synchronized BAT response to environmental challenges”. Neurovascular remodeling would also be “harmonized” if the same ligand acted on the vasculature and sympathetic fibers. What are the benefits of having a bifurcated system? A 1-2 sentence explanation in the Discussion would suffice.

2. The types of quantification should be consistent across all figures to facilitate comparisons between models. For example, Figure 6j shows the number of TH+ neurites/area while Figure 6l shows % TH staining. Either method is fine.

3. Figure 1 highlights the identification of Robo4 expression in vascular and lymphatic endothelial cells but not adipocytes. Subsequently, the authors reinforce this point by showing that brown adipocytes cannot respond to Slit3 in vitro (Suppl Fig 6a-b). It is then surprising that the authors do not show any data from Robo4 KD. The authors should consider removing Robo4-related data or at least de-emphasizing it in the initial rationale for the experiments, if they cannot report results of Robo4 KD.

Version 1:

Reviewer comments:

Reviewer #1

(Remarks to the Author)

Overall, the authors have addressed my concerns, and the manuscript is a valuable contribution to the field of brown adipose tissue biology. However, I feel they are making an inference error by not including normalized values. The author states:

“That said, we respectfully believe that normalizing to adipocyte area is not biologically appropriate in this context. Several lines of evidence presented in the manuscript indicate that the whitening phenotype is driven by reduced sympathetic innervation and vascular density, which secondarily influences adipocytes and leads to whitening of BAT. Normalizing to adipocyte area would obscure these causal relationships by treating change in adipocyte size as an independent confounder, when in fact it is a consequence of reduced sympathetic input.”

The problem here is that the authors are confusing causality with measurement validity. They are committing a causal direction fallacy, by assuming that because they understand the causal chain, they should measure in a way that reflects that chain, rather than measuring a parameter in an objective way.

The functional vascularization of the tissue is the measurement of vascular density per cell (or per unit of tissue cellularity).

The measurement question is independent of causality. Normalizing to adipocyte area doesn't claim adipocyte size is an independent confounder. It simply accounts for the fact that you're comparing tissues with different cellular densities. If anything, not normalizing obscures whether vascular changes are proportional to cellular changes or represent an additional effect.

The ideal solution is to pose the hypothesis (If sympathetic denervation → reduced vascularization → adipocyte enlargement), and provide measurements to answer two questions:

- Is vascular density per unit area reduced? (answered without normalization)
- Is vascular density per cell reduced? (requires normalization to cell number or area)

Reviewer #2

(Remarks to the Author)

The authors have satisfactorily addressed my previous concerns, except for the requested immunostaining for Slit3. This could not be completed due to the lack of a specific antibody, which is entirely understandable.

I appreciate that the authors added a concise section summarizing the Slit-Robo signaling pathway and its established roles in neurogenesis and angiogenesis in the Introduction. This addition provides valuable context for readers who may be less familiar with this signaling axis. However, citing the major original papers in this area would further strengthen the manuscript and assist readers who wish to explore the related literature in greater depth.

Reviewer #3

(Remarks to the Author)

The authors performed additional experiments and analyses that strengthened their central claims. Studies in KDs at thermoneutrality support critical roles for Slit3 both in establishing the BAT neurovascular network of BAT and in driving its remodeling during cold exposure. Moreover, assessment of Slit3 levels in cKOs in the cold supports the idea that APCs are the primary source of Slit3. Finally, they demonstrated that brown adipocyte responses to adrenergic stimulation do not require Slit3. These experiments, as well as additions to the discussion, fully addressed all my concerns.

Version 2:

Reviewer comments:

Reviewer #1

(Remarks to the Author)

The authors have addressed all my concerns.

REVIEWER COMMENTS

Reviewer #1 (Remarks to the Author):

In this manuscript, the authors identify Slit3 as a key factor mediating the neurovascular organization of brown adipose tissue (BAT) in mice. Using single-cell analyses and western blotting, they confirm that Slit3 is expressed in adipose tissue progenitor cells. They further demonstrate that Slit3 enhances vascularization and innervation of BAT in response to cold exposure and plays a major role in supporting thermogenesis under these conditions. The functional effects are attributed to the C-terminal fragment of Slit3, processed by BMP1 and signaling through Plxna1. Overall, the experiments are well described and the results largely support the authors' conclusions. However, certain aspects require additional evidence or clarification.

We thank the reviewer for their thoughtful assessment of our manuscript.

Specific Points:

1. The authors report that reduction of Slit3 leads to whitening of BAT under cold conditions but do not provide an explanation for this observation. Is BAT accumulating lipids released from lipolysis in other tissues? What are the weights of the inguinal and epididymal fat pads in these mice?

We thank the reviewer for this insightful question. We have included the weights of the inguinal and epididymal fat pads in Slit3 knockdown and control mice in the revised manuscript. As shown in Supplementary Figure 5b, there are no differences between the groups, indicating that whitening of BAT in Slit3 knockdown mice is not due to redistribution of lipids from white adipose tissue depots.

Instead, gene expression analysis reduced expression of the lipolytic enzymes Atgl (*Pnpla2*) and Hsl (*Lipe*) in BAT from Slit3 knockdown mice (Figure 1i). This indicates that loss of Slit3 directly reduces lipolysis within BAT itself, leading to higher lipid accumulation. In addition, BAT from Slit3 knockdown mice showed decreased expression of thermogenic and mitochondrial genes and proteins. Together, these findings indicate that the whitening phenotype arises from a combination of reduced lipolytic capacity and impaired mitochondrial oxidation in BAT of Slit3 loss-of-function mice.

2. Angiogenesis is assessed via isolectin staining. However, because adipocytes enlarge under certain conditions, vessel density per field can appear reduced. This must be controlled for by normalizing to adipocyte density—for example, expressing isolectin intensity relative to LipiTox or PLIN1 staining intensity. The same normalization should apply to TH staining.

We thank the reviewer for raising this point. We agree that adipocyte hypertrophy can influence the apparent density of vessels or sympathetic fibers within a given field. To address this concern, we normalized the IB4- and TH-positive areas to the Plin1-positive area. Notably, even after this normalization, the relative densities of IB4⁺ capillary vessels and TH⁺ sympathetic fibers remained significantly reduced in shSlit3 BAT, confirming that the observed effect is not simply an artifact of increased adipocyte size (Figure 1 in the point-by-point document).

Figure 1. Quantification of IB4⁺ (left) and TH⁺ (right) areas normalized to the Plin1⁺ (adipocyte) area in BAT from mice treated with AAV-shSlit3 or scramble shRNA after 7 days of cold exposure (5 °C). Data are shown as mean ± SEM and were analyzed using unpaired two-sided Student's t-tests.

That said, we respectfully believe that normalizing to adipocyte area is not biologically appropriate in this context. Several lines of evidence presented in the manuscript indicate that the whitening phenotype is driven by reduced sympathetic innervation and vascular density, which secondarily influences adipocytes and leads to whitening of BAT. Normalizing to adipocyte area would obscure these causal relationships by treating change in adipocyte size as an independent confounder, when in fact it is a consequence of reduced sympathetic input. Importantly, total BAT mass does not change between the groups (Supplementary Figure 5b). For these reasons, we consider field-based quantification the most physiologically relevant approach in this case.

3. In shRNA-treated mice, BAT shows a marked increase in lipid droplet size. Is this phenotype recapitulated in the Pdgfra-Cre-driven Slit3 knockout (Fig. 3)? If so, isolectin and TH staining data must also be corrected for adipocyte density, as recommended for Fig. 2.

H&E analysis of the BAT from Pdgfra-creERT2;Slit3^{flox/flox} mice did not reveal a difference in lipid droplet size. These results have been included in the revised manuscript (Figure 3k). Accordingly, capillary density and sympathetic innervation are presented as a percentage of the total tissue area.

4. Enforced AAV-mediated expression of the C-terminal or full-length Slit3 increases isolectin staining, suggesting enhanced angiogenesis (Fig. 5). However, adipocyte density should be reported, and the data normalized

accordingly. The rationale for performing these experiments under conditions where endogenous Slit3 expression is already expected to be high (WT mice after 7 days of cold exposure) is unclear. A more compelling experiment would assess the effect of Slit3-C in non-cold-exposed mice, where any increase in innervation could be attributed directly to the construct.

We thank the reviewer for raising these points. We would like to clarify that overexpression of full-length Slit3 or the Slit3-N fragment, but not Slit3-C, resulted in increased isolectin B4 staining and angiogenesis. H&E analysis of BAT from mice overexpressing Slit3-FL, Slit3-N, or Slit3-C revealed no differences in adipocyte size. Therefore, to ensure consistency with the loss of function studies, we quantified both TH and IB4 staining as a percentage of the total tissue area.

To address the reviewer's comment regarding the effects of Slit3 overexpression at room temperature, we have included new data from mice overexpressing full-length Slit3, Slit3-N, and Slit3-C and housed at room temperature (Supplementary Figure 11). Consistent with the results observed under cold conditions, overexpression of Slit3-FL and Slit3-C, but not Slit3-N, increased TH levels in BAT (Supplementary Figure 11). These effects were, however, more modest at room temperature compared to cold condition. Moreover, overexpression of Slit3-FL and Slit3-C at room temperature did not increase UCP1 protein levels, indicating that while Slit3-FL and Slit3-C can directly promote sympathetic innervation under basal conditions, their impact becomes more pronounced during cold exposure, when sympathetic activation and thermogenic demand are elevated (Supplementary Figure 11). This observation is also consistent with our loss-of-function studies, which show that although basal sympathetic innervation is reduced in Slit3-deficient BAT at room temperature, cold exposure further amplifies this deficit and its impact on BAT thermogenesis (Figures 1-2 and Supplementary Figures 4 and 7).

5. The hypothesis that BMP1 functions as a protease for Slit3 is supported by data showing diminished proteolysis following siRNA-mediated Bmp1 knockdown or pharmacological inhibition. While these findings suggest a requirement for BMP1 in Slit3 cleavage, they do not definitively establish BMP1 as the responsible protease. Rigorous molecular characterization—including specificity and kinetic studies—would be required to confirm this. Similarly, while structural modeling and shRNA knockdown data support the role of Plxna1 as a receptor for Slit3-C, definitive evidence requires demonstration of direct receptor-ligand binding, including affinity and specificity measurements. Although these experiments may be beyond the scope of the current study, the limitations of the conclusions should be clearly acknowledged in the manuscript.

We appreciate the reviewer's thoughtful feedback. We agree that detailed molecular characterization of protease activity and affinity or specificity measurements would provide valuable mechanistic insight. However, as the reviewer points out, these analyses are beyond the scope of the present study, which focuses on defining the physiological roles of Slit3 fragments in BAT. Regarding the Slit3-Plxna1 interaction, we present three independent lines of

evidence, co-immunoprecipitation, structural modeling, and in vivo data, all supporting Plxna1 as a bona fide receptor for Slit3-C.

In line with the reviewer's suggestion, we have added a statement to the manuscript acknowledging these limitations and outlining potential directions for future investigation.

Reviewer #2 (Remarks to the Author):

General Assessment:

This study investigates the molecular mechanisms coordinating neurovascular expansion and thermogenesis in brown adipose tissue (BAT), with a focus on the role of adipocyte progenitors. Building on previous scRNA-seq findings that identified adipocyte progenitors as key communication hubs, the authors perform in-depth bioinformatics analyses to identify secreted ligands mediating intercellular crosstalk. They identify secreted axon guidance molecule Slit3 and demonstrate through functional experiments that Slit3 is essential for BAT thermogenesis, angiogenesis, and sympathetic innervation.

The study further reveals that Slit3 is proteolytically processed by Bmp1 into distinct fragments (Slit3-FL, Slit3-N, and Slit3-C), each with specific roles in neurovascular remodeling. AAV-mediated gain-of-function experiments show that Slit3-C promotes sympathetic innervation, while Slit3-N enhances angiogenesis. The authors also identify Plxna1 as a receptor mediating Slit3-C-dependent sympathetic innervation. Finally, the study links Slit3 expression to adipose tissue health and inflammation in humans.

This is a well-structured and compelling study that provides novel mechanistic insights into how adipocyte progenitors orchestrate neurovascular remodeling in BAT. The findings are timely and significantly advance our understanding BAT biology, and the experimental design is generally robust. While there are no major concerns, the following suggestions may help further strengthen the manuscript:

We thank the reviewer for their thoughtful assessment of our manuscript.

Specific Comments:

1. Figure 1C: The increase in Slit3-FL under cold conditions is shown, but it is unclear whether similar changes occur for Slit3-N or Slit3-C. If the antibody used is expected to recognize Slit3-N or Slit3-C, this should be clarified. Including this information as supplementary data would enhance transparency.

The antibody used in our study (AF3629, R&D Systems) recognizes the N-terminal region of the Slit3 protein, and we have revised the manuscript text to clarify this point. However, this antibody does not reliably detect Slit3 fragments in tissue samples. We systematically evaluated all commercially available Slit3 antibodies and generated two custom antibodies; however, none demonstrated sufficient specificity for reliable use. Therefore, we are currently unable to

accurately assess changes in Slit3-N or Slit3-C levels under cold conditions due to this technical limitation.

2. Supplementary Figure 2a–d: To support the claim that Slit3 is not expressed in macrophages under basal or cold conditions, immunofluorescence staining could provide additional confirmation.

We agree that immunofluorescence staining would provide valuable confirmation of Slit3 expression patterns. However, to date there are no reliable antibodies available for Slit3 immunofluorescence. We have tested several commercially available antibodies and even attempted to generate new ones, but unfortunately none produced specific or reproducible immunofluorescence signal.

3. Supplementary Figure 2e: The use of a mouse adipocyte progenitor cell line to assess Slit3 expression during adipogenesis is informative. However, staining for Slit3 in both mature adipocytes and progenitors in BAT and WAT would provide more comprehensive validation.

Unfortunately, as noted above, the lack of reliable antibodies for Slit3 immunofluorescence has prevented us from performing such experiments. We tested multiple commercially available antibodies and attempted to generate new ones, but none were effective for IF or IHC applications. Despite this limitation, the concordance of our mRNA expression data from single-cell transcriptomics, FACS-isolated primary cells, and in vitro-differentiated adipocyte progenitors strongly supports the validity of the conclusions.

4. Figure 2g: A protein loading control is missing. Including this would strengthen the interpretation of the immunoblot data.

We have added a protein loading control and quantified TH levels relative to total protein to ensure accurate normalization.

5. Figure 3a: Similarly, a loading control is needed. Given that Slit3 is expressed in multiple cell types in BAT, and that Slit3 expression is expected to be reduced but not absent in Slit3iDAPC tissue, it would be helpful to show Slit3 protein levels under both basal and cold conditions with appropriate controls.

We have incorporated new data quantifying Slit3 protein levels in BAT from *Pdgfra-creERT2;Slit3^{flox/flox}* mice under both room temperature and cold conditions, as well as in ingWAT and pgWAT. Loading controls and total protein stainings are included (Figure 3a-h and Supplementary Data 8). These results reveal a significant reduction in Slit3 protein levels in BAT under room temperature and cold conditions, with the decrease being more pronounced after 7 days of cold exposure (Figure 3a-d).

6. Figure 5g–h: Overexpression of Slit3-C also increases capillary density in BAT.

This is unexpected given the fragment's primary role in sympathetic innervation. A brief discussion of potential mechanisms would be valuable.

We thank the reviewer for raising this point. The previous version of the manuscript had a typo in the description of the results in Figure 5g-h. As correctly shown in the Figure 5g-h, overexpression of Slit3-C **does not** increase capillary density in BAT (the purple vs grey bars). This typo has now been corrected in the revised version.

7. Figure 6d: The lack of co-immunoprecipitation between Plxna1 and Slit3-N supports the specificity of Plxna1 for Slit3-C. This point could be emphasized more clearly to strengthen the conclusion.

Since overexpression of Slit3-N did not affect sympathetic innervation in BAT, and Plxna1 expression was detected specifically on sympathetic neurites, we do not anticipate a direct or physiologically relevant interaction between Plxna1 and Slit3-N in this context. Nevertheless, to explore this possibility *in silico*, we employed AlphaFold2 Multimer to model complexes between the extracellular domain of Plxna1 and the N-terminal region of Slit3. The resulting model predicted potential contacts between several Leucine-rich repeat (LRR) domains of Slit3-N and Plxna1. However, this predicted interaction is unlikely to represent a bona fide signaling event and may instead arise from the characteristic solenoid architecture of LRR domains, which commonly serve as structural scaffolds mediating generic protein-protein contacts rather than specific ligand-receptor interactions¹.

8. Figure 1i: The p-value for Dio2 is 0.062. While not statistically significant, the trend may still be biologically relevant and could be briefly discussed.

We have now mentioned Dio2 and other thermogenic genes in the manuscript text.

9. Line 291: There appears to be a typo—"Slit3-FL" should be "Slit3-C."

As shown in Figure 5, both Slit3-FL and Slit3-C significantly enhance sympathetic innervation in BAT. The section referenced by the reviewer discusses the effect of Slit3-FL overexpression in the presence or absence of Plxna1. Similar results for Slit3-C overexpression are described later in the text.

10. Supplementary Figure 6e: The high-magnification images do not appear to align with the low-magnification views. Please verify and adjust if necessary.

We thank the reviewer for identifying the error in the figure. The high-magnification images for the two groups have been corrected and appropriately swapped in the revised version.

11. Literature Context: Including a brief section summarizing the Slit-Robo signaling pathway and its known roles in neurogenesis and angiogenesis in the Introduction would provide helpful context for readers less familiar with this signaling axis.

We have now added a brief description of the Slit-Robo signaling pathway and its roles in neurogenesis, angiogenesis, and other processes to the Introduction.

Reviewer #3 (Remarks to the Author):

Cold exposure leads to a marked increase in vascularization and sympathetic nervous system (SNS) innervation of thermogenic adipose tissue to meet increased energetic demands. Several independent groups implicated neurotrophins and Slit3 secreted by myeloid cells in the cold-induced expansion of sympathetic innervation in inguinal white adipose tissue (iWAT) (PMID: 33297933, PMID: 35042776; PMID: 29589323; PMID: 34782792). This manuscript identifies key molecular mediators of cold-induced neurovascular expansion in brown adipose tissue (BAT). First, the authors demonstrate that Slit3 is predominantly expressed in adipocyte progenitor cells (APCs) and vascular smooth muscle in BAT, and not myeloid cells as in iWAT. Complementary loss-of-function studies using AVV-mediated expression of shRNAs in BAT and genetic knockout in APCs provide compelling evidence that Slit3 is necessary for cold-induced increases in angiogenesis and sympathetic innervation in BAT. In vitro loss-of-function and in vivo gain of function studies support the idea that the metalloprotease BMP1 cleaves Slit3 into two fragments that regulate distinct functions. The N-terminal fragment promotes angiogenesis (consistent with PMID: 22641771), while the C-terminal fragment increases SNS innervation. The authors identified Plxn1 as a potential receptor for the C-terminal fragment of Slit3 using computational modeling. Complementary in vitro and in vivo studies support the idea that Plxn1 is a key receptor mediating the effects of the full length or C-terminal fragment of Slit3 on SNS innervation in BAT. Expression studies identified Robo4 as a putative receptor on endothelial cells, but its requirement for cold-induced angiogenesis was not tested.

In summary, while Slit3 appears to regulate similar adaptive responses to cold in iWAT and BAT, these studies show that the source of Slit3 (myeloid cells vs APCs) and the specific receptors mediating effects on the SNS and vasculature are distinct (Robo1 vs Plxn1). The novelty of these studies lies in the demonstration that BMP1 cleaves Slit3 into 2 fragments that act on the SNS and vasculature via distinct receptors. The authors propose that the distinct activities of Slit3 proteolytic cleavage products provide a means to synchronize adaptive

responses to cold. Overall, the complementary and rigorous approaches provide very strong support for the molecular pathways outlined here. However, there are several outstanding issues that need to be resolved, outlined below.

We thank the reviewer for their thoughtful assessment of our manuscript.

1 The most fundamental outstanding issue is whether Slit3 only functions in the context of cold-induced remodeling or also maintains the neurovascular architecture at thermoneutrality (i.e. baseline). This is critical to know if you want to translate these findings to humans, who live near thermoneutrality. Slit3 KD reduced neurovascular density at room temperature (Suppl Fig 7), which is a mild cold challenge. Analyzing Slit3 KD or cKO models at thermoneutrality would be informative as to whether Slit3's function is restricted to cold-induced expansion or also affects baseline structure.

We thank the reviewer for raising this point. To address it, we conducted additional experiments in which mice injected with Slit3 shRNAs or scramble control were housed under thermoneutral conditions (30 °C) for two weeks. Mice lacking Slit3 expression in BAT showed a comparable reduction in Ucp1 protein at thermoneutrality (Supplementary Figure 4d-f). The expression of other thermogenic and mitochondrial genes was also modestly reduced under these conditions (Supplementary Figure 4d). Notably, even when animals were maintained at thermoneutrality, loss of Slit3 led to a decrease in both capillary density and sympathetic innervation in BAT (Supplementary Figure 7e-h). These findings demonstrate the essential role of Slit3 in both establishing the neurovascular network of BAT under basal conditions and driving its expansion and remodeling during cold adaptation.

2. Whereas APCs express the highest levels of Slit3, cold-induced increases in transcription are only observed in ECs (Suppl Fig 1d). Moreover, AAV-mediated Slit3 KD in BAT (Fig. 1e) produced a stronger reduction in body temperature after 5 hr at 5C than conditional KO in Pdgfr+ APCs (Figure 3d) (KD = 33.5C vs KO= 36.5C). This raises the possibility that adipocyte-derived Slit3 helps to maintain the neurovascular architecture at baseline but that EC-derived Slit3 mediates cold-induced remodeling. Comparisons between adipocyte-and EC-specific Slit3 KOs at thermoneutrality and in response to a cold challenge would resolve this issue as well as issue #1.

We thank the reviewer for these comments. To evaluate the contribution of Pdgfra⁺ APCs to total Slit3 protein levels in BAT under basal and cold conditions, we have now quantified Slit3 protein in BAT from Pdgfra-creERT2;Slit3^{flox/flox} mice housed at room temperature or exposed to cold (Figure 3a-d). These data reveal a marked reduction in Slit3 protein under both conditions, with a more pronounced decrease observed following 7 days of cold exposure. This

reduction, together with the significant loss of capillary density and sympathetic innervation in BAT from cold-exposed *Pdgfra-creERT2;Slit3^{flox/flox}* mice, underscores the essential role of *Pdgfra*⁺ APC-derived Slit3 in driving cold-induced neurovascular remodeling.

The differences in body temperature observed between the *Pdgfra-creERT2;Slit3^{flox/flox}* and Slit3 knockdown models may reflect variations in experimental design, mouse strain, and genetic background. While we agree that generating and comparing an EC-specific Slit3 knockout model would provide additional insight, this lies beyond the scope of the present study, which focuses on defining the novel role of *Pdgfra*⁺ APC-derived Slit3, its proteolytic processing, and the distinct functions of its cleavage fragments through different receptors in coordinating the expansion of the BAT neurovascular network.

We have added a limitations section to the revised manuscript, in which we discuss the study's limitations and acknowledge the potential contribution of EC-derived Slit3 to adipose tissue function and remodeling.

3. While the data supporting a role for Slit3 in cold-induced neurovascular remodeling are compelling, the relationship with BAT thermogenesis is less clear. In the Slit3 KD, BAT temperature (Suppl Fig 3e) and body temperature (Fig 1e) are lower in the first 5 hours of cold exposure – this is before any cold-induced remodeling occurs. On the other hand, after 2 days of cold exposure, when Slit3 levels are high (Figure 1d), BAT temperature of KDs is the same as controls (Suppl Fig 3e). Additionally, *Plxna1* KD in conjunction with Slit3-C overexpression prevents the cold-induced increase in SNS innervation (i.e. innervation looks like controls), but the BAT temperature curve is similar to that of the *Plxna1* KD alone, which has lower SNS innervation (Figure 6 l,m). Unless these apparent inconsistencies can be definitively resolved, I would recommend removing the body/BAT temperature data and narrowing the claims to SNS and vascular remodeling.

We thank the reviewer for these insightful comments. While it is true that body temperature in Slit3 KD animals eventually normalizes after the first ~12 hours of cold exposure, this likely reflects the activation of compensatory thermoregulatory mechanisms such as shivering, which allow animals to survive chronic cold stress. Importantly, our molecular analyses show that Slit3 KD animals exhibit significant impairments in the activation of thermogenic gene and protein program in BAT after 7 days of cold exposure (Figure 1f-k). They also exhibit lower energy expenditure (Figure 1l-m), oxygen consumption, and carbon dioxide production when housed chronically in cold (Supplementary Figure 5). Collectively, these findings indicate that, despite the recovery of body temperature, loss of Slit3 disrupts BAT thermogenic capacity.

Furthermore, to directly test whether impaired sympathetic innervation underlies the thermogenic defects in Slit3-deficient BAT, we designed an experiment to bypass SNS input by administering the β 3-adrenergic receptor agonist CL-316,243. Slit3 knockdown animals treated with CL-316,243 for ten days exhibited fully intact induction of thermogenic, mitochondrial, and lipolytic gene programs (Supplementary Figure 7i). These results demonstrate that the intrinsic ability of brown adipocytes to respond to adrenergic stimulation remains uncompromised in the absence of Slit3. Thus, the primary role of Slit3 in BAT is not to modulate adipocyte-intrinsic thermogenic capacity, but rather to drive the proper neurovascular expansion necessary for effective cold-induced thermogenesis. We have incorporated these new data and their description into the revised manuscript.

Regarding the data in Figure 6l-m, we agree with the reviewer about the partial disconnect between sympathetic innervation levels and BAT temperature. We speculate that this may be due to additional roles of Plxna1 in sympathetic neurons beyond axon growth, potentially involving pathways downstream of semaphorin ligands. We have expanded the discussion to address the possible contribution of non-Slit3 ligands to Plxna1 signaling and their relevance to thermogenic regulation. While delineating the relative contribution of different ligands is beyond the scope of the current study, we recognize that Plxna1 likely has Slit3-independent roles in BAT physiology. We have revised the manuscript accordingly to clarify these points.

4. Analyses of mice focused on the impacts of the transition from a moderate (22C) to a severe (5C) cold challenge on endpoints related to neurovascular remodeling in lean animals. In contrast, samples of abdominal subcutaneous and visceral omental WAT depots were obtained from obese people living near thermoneutrality. The most compelling associations are a positive correlation between Slit3 and adiponectin expression in abdominal subcutaneous WAT and a negative correlation with macrophages in omental visceral WAT (which could be due to the increase in adiponectin). Analysis of markers of SNS tone (i.e. TH) or capillaries would help to connect Figure 8 to the rest of the manuscript. Otherwise, the manuscript would be strengthened by saving these interesting human data for another manuscript.

We acknowledge the limitations of the human studies, including the differences in physiological state and depot differences due to the difficulties of accessing fresh BAT biopsies, which precludes direct measurement of sympathetic innervation or capillary density. However, we believe these data provide important complementary insight. Specifically, the observed associations between Slit3 expression and markers of adipose tissue health such as increased adiponectin and decreased macrophage abundance offer the first evidence that the Slit3 pathway may be relevant in regulation of human adipose tissue function and health. While these data are correlative and do not establish causality, we

feel that they are biologically meaningful and add translational relevance to the mouse findings. We have revised the manuscript to better clarify these limitations and to more explicitly frame the human data as hypothesis-generating. We hope the reviewer agrees that including these results strengthens the manuscript by broadening its scope and potential impact.

Minor Concerns:

1. The authors posit that “the co-regulation of neurovascular expansion by distinct Slit3 fragments offers a bifurcated yet harmonized mechanism to ensure a synchronized BAT response to environmental challenges”. Neurovascular remodeling would also be “harmonized” if the same ligand acted on the vasculature and sympathetic fibers. What are the benefits of having a bifurcated system? A 1-2 sentence explanation in the Discussion would suffice.

We have amended to the Discussion to describe the potential benefits of a bifurcated signaling system. Specifically, we note that a bifurcated signaling system enables independent yet coordinated regulation of vascular and neuronal components, providing greater flexibility and temporal precision in adapting to environmental stimuli. In the context of Slit3, proteolytic processing introduces an additional level of regulation, as the full-length protein and its fragments exhibit distinct cell association and diffusion properties². Notably, Slit3-C is exclusively detected in the media and not in the cell lysate, suggesting enhanced diffusibility and a capacity to mediate long-range effects on sympathetic innervation, whereas Slit3-FL and Slit3-N are more likely to function in a localized manner (Figure 4c and reference 2).

2. The types of quantification should be consistent across all figures to facilitate comparisons between models. For example, Figure 6j shows the number of TH+ neurites/area while Figure 6l shows % TH staining. Either method is fine.

We thank the reviewer for this suggestion. To ensure consistency and facilitate comparisons between models, we have revised all quantifications to present data as the percentage of TH⁺ or IB4⁺ areas across all figures.

3. Figure 1 highlights the identification of Robo4 expression in vascular and lymphatic endothelial cells but not adipocytes. Subsequently, the authors reinforce this point by showing that brown adipocytes cannot respond to Slit3 in vitro (Suppl Fig 6a-b). It is then surprising that the authors do not show any data from Robo4 KD. The authors should consider removing Robo4-related data or at least de-emphasizing it in the initial rationale for the experiments, if they cannot report results of Robo4 KD.

We thank the reviewer for this comment and suggestion. Ligand-receptor analysis using single-cell transcriptomic data identified the Slit3-Robo4 pair, which provides the rationale for investigating the role of Slit3 in BAT. We believe including this rationale helps readers understand the logic of the study. As prior work has established that Robo4 mediates the angiogenic effects of Slit family members through binding to the N-terminal region, in this study we focused on elucidating the novel Slit3C-Plxna1 interaction and its specific role in regulating sympathetic innervation within BAT. Collectively, our findings, together with prior literature, reveal distinct and complementary mechanisms of Slit3 signaling that enables coordinated regulation of neurovascular expansion and remodeling.

References

1. Kobe, B. & Kajava, A. V. The leucine-rich repeat as a protein recognition motif. *Curr. Opin. Struct. Biol.* **11**, 725–732 (2001).
2. Brose, K. *et al.* Slit proteins bind Robo receptors and have an evolutionarily conserved role in repulsive axon guidance. *Cell* **96**, 795–806 (1999).

REVIEWER COMMENTS

Reviewer #1 (Remarks to the Author):

Overall, the authors have addressed my concerns, and the manuscript is a valuable contribution to the field of brown adipose tissue biology. However, I feel they are making an inference error by not including normalized values. The author states:

“That said, we respectfully believe that normalizing to adipocyte area is not biologically appropriate in this context. Several lines of evidence presented in the manuscript indicate that the whitening phenotype is driven by reduced sympathetic innervation and vascular density, which secondarily influences adipocytes and leads to whitening of BAT. Normalizing to adipocyte area would obscure these causal relationships by treating change in adipocyte size as an independent confounder, when in fact it is a consequence of reduced sympathetic input.”

The problem here is that the authors are confusing causality with measurement validity. They are committing a causal direction fallacy, by assuming that because they understand the causal chain, they should measure in a way that reflects that chain, rather than measuring a parameter in an objective way. The functional vascularization of the tissue is the measurement of vascular density per cell (or per unit of tissue cellularity). The measurement question is independent of causality. Normalizing to adipocyte area doesn't claim adipocyte size is an independent confounder. It simply accounts for the fact that you're comparing tissues with different cellular densities. If anything, not normalizing obscures whether vascular changes are proportional to cellular changes or represent an additional effect.

The ideal solution is to pose the hypothesis (If sympathetic denervation → reduced vascularization → adipocyte enlargement), and provide measurements to answer two questions:

- Is vascular density per unit area reduced? (answered without normalization)
- Is vascular density per cell reduced? (requires normalization to cell number or area)

We thank the reviewer for their positive evaluation and for their careful consideration of our study. In response to the reviewer's comment, we have added new graphs that explicitly shows both tissue-level and adipocyte-normalized vascular and sympathetic measurements, as suggested. Specifically, we now present sympathetic innervation and vascular density quantified in two ways: normalized to total tissue area and normalized to adipocyte area (Perilipin1-positive area). These additional analyses are included for all shSlit3 studies (Figure 2b-c, Figure 2f-g, and Supplementary Figure 7), for experiments

involving Slit3 fragment overexpression (Figure 5e-f and Figure 5h), and for comparisons of vascular and innervation density in wild-type mice at different temperatures (Supplementary Figure 6). This dual normalization directly allows assessment of (i) changes in vascular and sympathetic density per unit tissue area and (ii) changes relative to adipocyte content, addressing whether vascular alterations are proportional to cellular changes or represent an additional effect. Collectively, these results show that the reduced sympathetic innervation and vascular density are not secondary to adipocyte enlargement.

For the *Plxna1* knockdown studies (Figures 6-7), *Perilipin1* co-staining was not performed in the original experiments. Importantly, these conditions show no changes in BAT morphology or adipocyte size. After consultation with the editor, we therefore retained the original quantification normalized to total tissue area, as additional normalization would not alter interpretation in the absence of adipocyte size differences.

In summary, across all conditions, normalization to adipocyte area does not change the direction or significance of the observed effects, and the conclusions of the study remain unchanged. To ensure full transparency, we also explicitly state the quantification and normalization methods used in each figure legend.

Reviewer #2 (Remarks to the Author):

The authors have satisfactorily addressed my previous concerns, except for the requested immunostaining for Slit3. This could not be completed due to the lack of a specific antibody, which is entirely understandable.

I appreciate that the authors added a concise section summarizing the Slit-Robo signaling pathway and its established roles in neurogenesis and angiogenesis in the Introduction. This addition provides valuable context for readers who may be less familiar with this signaling axis. However, citing the major original papers in this area would further strengthen the manuscript and assist readers who wish to explore the related literature in greater depth.

We thank the reviewer for their positive evaluation and for their careful consideration of our study. We have now included additional references citing key papers and discoveries related to Slit-Robo signaling and its role in tissue development and remodeling.

Reviewer #3 (Remarks to the Author):

The authors performed additional experiments and analyses that strengthened their central claims. Studies in KDs at thermoneutrality support critical roles for Slit3 both in establishing the BAT neurovascular network of BAT and in driving its

remodeling during cold exposure. Moreover, assessment of Slit3 levels in cKOs in the cold supports the idea that APCs are the primary source of Slit3. Finally, they demonstrated that brown adipocyte responses to adrenergic stimulation do not require Slit3. These experiments, as well as additions to the discussion, fully addressed all my concerns.

We thank the reviewer for their positive evaluation and for their careful consideration of our study.